

# Aerosol backscatter profiles from ceilometers: validation of water vapor correction in the framework of CeiLinEx2015

Matthias Wiegner[1], Ina Mattis[2], Margit Pattantyús-Ábrahám[2,a], Juan Antonio Bravo-Aranda[3,b],
Yann Poltera[4,c], Alexander Haefele[4], Maxime Hervo[4], Ulrich Görsdorf[5], Ronny Leinweber[5],
Josef Gasteiger[6], Martial Haeffelin[3], Frank Wagner[2,d], Jan Cermak[7,e,f], Katerina Komínková[8],
Mike Brettle[9], Christoph Münkel[10], and Kornelia Pönitz[11]

[1]Meteorologisches Institut, Ludwig-Maximilians-Universität, Theresienstraße 37, 80333 München, Germany
[2]Deutscher Wetterdienst, Meteorologisches Observatorium Hohenpeißenberg, Hohenpeißenberg, Germany
[3]Institut Pierre Simon Laplace, École Polytechnique, CNRS, Université Paris–Saclay, Palaiseau, France
[4]MeteoSwiss, Payerne, Switzerland
[5]Deutscher Wetterdienst, Meteorologisches Observatorium Lindenberg, Lindenberg, Germany
[6]Faculty of Physics, University of Vienna, Vienna, Austria
[7]Department of Geography, Ruhr-Universität Bochum, Bochum, Germany
[8]Global Change Research Institute, Czech Academy of Sciences, Brno, Czech Republic
[9]Chartered meteorologist, UK
[10]Vaisala GmbH, Hamburg, Germany
[11]G. Lufft Mess- und Regeltechnik GmbH, Fellbach, Germany
[a]now at: Federal Office for Radiation Protection, Department of Environmental Radiation, Neuherberg, Germany
[b]now at: University of Granada, Granada, Spain
[c]now at: Institute for Atmospheric and Climate Science, ETH Zurich, Zurich, Switzerland
[d]now at: Karlsruhe Institute of Technology (KIT), IMK–TRO, Eggenstein-Leopoldshafen, Germany
[e]now at: Karlsruhe Institute of Technology (KIT), Institute of Meteorology and Climate Research, Karlsruhe, Germany
[f]now at: Karlsruhe Institute of Technology (KIT), Institute of Photogrammetry and Remote Sensing, Karlsruhe, Germany

**Correspondence:** Matthias Wiegner (m.wiegner@lmu.de)

**Abstract.** With the rapidly growing number of automated single-wavelength backscatter lidars (ceilometers) their potential benefit for aerosol remote sensing received considerable scientific attention. When studying the accuracy of retrieved particle backscatter coefficients it must be considered that most of the ceilometers are influenced by water vapor absorption in the spectral range around 910 nm. In the literature methodologies to correct for this effect have been proposed, however, a val-

5  idation was not yet performed. In the framework of the ceilometer intercomparison campaign CeiLinEx2015 in Lindenberg, Germany, hosted by the German Weather Service, it was possible to tackle this open issue. Ceilometers from Lufft (CHM15k and CHM15kx, operating at 1064 nm), from Vaisala (CL51 and CL31) and from Campbell Scientific (CS135), all operating at a wavelength of approximately 910 nm, were deployed together with a multi-wavelength research lidar (RALPH) that served as reference. In this paper the validation of the water vapor correction is performed by comparing ceilometer backscatter signals with measurements of the reference system extrapolated to the water vapor regime. One inherent problem of the

10  validation is the spectral extrapolation of particle optical properties. For this purpose AERONET measurements and inversions of RALPH signals were used. Another issue is that the vertical range where validation is possible is limited to the upper part of the mixing layer due incomplete overlap, and the in general low signal to noise ratio and signal artefacts above that layer.



Our intercomparisons show that the water vapor correction leads to a quite good agreement between the extrapolated reference signals and the measurements in the case of CL51 ceilometers at one or more wavelengths in the specified range of the laser diode's emission. This equivocation is due to the similar effective water vapor transmission at several wavelengths. In the case of CL31 and CS135 ceilometers the validation was not always successful. That suggests that error sources beyond the water vapor absorption might be dominant. For future applications we recommend to monitor the emitted wavelength and to provide "dark" measurements on a regular basis.

## 1 Introduction

In the last years a significant number of eye-safe single-wavelength backscatter lidars, so called ceilometers, has been installed for unattended operation. The primary reason to install ceilometer networks is the automation of synoptic observations, especially for the accurate determination of the cloud base height, but since approximately 2010 aerosol and ash remote sensing is considered as an additional application. Though aerosols are relevant for radiative transfer, cloud physics and air quality, the main driver of this application was the need for surveillance of the airspace in the case of a volcanic eruption. The Eyjafjallajökull event in 2010 and the subsequent restrictions for civil aviation impressively demonstrated the benefit of ceilometers (e.g., Flentje et al., 2010; Wiegner et al., 2012). In parallel efforts have been strengthened to derive not only mixing layer heights (e.g., Eresmaa et al., 2006; Münkel et al., 2007; Haeffelin et al., 2011; Lotteraner and Piringer, 2016; Geiß et al., 2017; Kotthaus and Grimmond, 2018a) but also optical properties, primarily profiles of the particle backscatter coefficient $\beta_p$ in a quantitative way. Recently ceilometer data were used for the validation of transport models, e.g. to improve forecasts of the dispersion of aerosol layers (e.g., Emeis et al., 2011; Cazorla et al., 2017; Chan et al., 2018), and to support air quality studies (e.g., Schäfer et al., 2011; Geiß et al., 2017; Kotthaus and Grimmond, 2018b), whereas data assimilation in numerical weather forecast models is still limited to case studies (e.g., Geisinger, 2017; Warren et al., 2018).

European ceilometer networks can be of particular benefit for the above mentioned purposes, when the spatiotemporal distribution of optical properties of particles is assessed in near real time. This requires a fully automated procedure of e.g. quality control, calibration, overlap correction, cloud clearing and more. Accordingly a huge research and development effort was coordinated in the framework of the COST-Action TOPROF (Towards operational ground based profiling with ceilometers, Doppler lidars and microwave radiometers for improving weather forecasts, Illingworth et al. (2018)) to make ceilometers exploitable for aerosol and ash profiling. E-PROFILE as part of the European Meteorological Services Network (EUMETNET) Composite Observing System (EUCOS) was established to integrate the ceilometers in Europe into an operational network, and to provide hand-in-hand with TOPROF data in real-time, well calibrated and quality controlled. E-PROFILE's key activity is an operational data hub, which collects, processes and redistributes ceilometer data. The scientific code run on the hub has been developed in TOPROF.

The development of such a processing chain is complicated because national operators rely on automated (low power) lidars and ceilometers (often referred to as ALC) from different manufacturers. For example, the German Weather Service has installed CHM15k-ceilometers (Lufft), whereas France, Finland and Switzerland rely on CL31 (Vaisala) for cloud detection.



Sweden has set up a network of CL31 ceilometer and only uses CBME-80 ceilometers (Eliasson) on airports. In UK both Lufft and Vaisala ceilometers are in operation. Compact micro-pulse lidars (MiniMPL) are used by Météo-France for volcanic ash detection, but also a limited set of advanced lidar systems are deployed, e.g. Polly$^{XT}$ in Finland and Raymetrics systems in UK. Recently, measurements of a CYY-2B ceilometer (CAMA) that was deployed in China were reported.

The ultimate goal of ceilometer measurements with respect to the quantitative retrieval of the aerosol optical properties is the provision of the particle backscatter coefficient $\beta_p(z)$ (Wiegner and Geiß, 2012). In this context the wavelength of the ceilometer is relevant: the above mentioned instruments operate either at 1064 nm (Lufft) or near 910 nm (Vaisala, Eliasson, Campbell, CAMA). The latter spectral range is influenced by water vapor absorption. As a consequence it is only possible to determine aerosol optical properties with additional knowledge of the water vapor distribution and properties of the ceilometers'

radiation source, and with the application of a correction scheme. Only if $\beta_p$ is derived with the best possible accuracy it might be used for estimates of further quantities (extinction coefficient, mass concentration), keeping in mind that the resulting accuracy is (drastically) reduced according to the accuracy of the inherent assumptions.

Even though water vapor absorption in the near infrared is well known it was often ignored. Sundström et al. (2009) evaluated CL31 measurements from 2005 in Helsinki, when they assumed that absorption of water vapor could be neglected. The same

assumption was made by Jin et al. (2015) using CL51 data. Comparisons of CL51 and CYY-2B measurements in Bejing, China, were also conducted without water vapor correction (Liu et al., 2018). Madonna et al. (2015) compared ceilometers of Lufft (CHM15k), Vaisala (CT25k) and Campbell (CS135s) in the framework of INTERACT (Potenza, Italy) but did not consider water vapor absorption quantitatively. To our knowledge, Markowicz et al. (2008) were the first who applied a correction term for water vapor absorption to data of a Vaisala CT25k-ceilometer before deriving aerosol optical properties. Wiegner et al.

(2014) discussed the problem in a general way on the basis of simulated signals and proposed an improved approach to correct for water vapor absorption. A follow-on paper (Wiegner and Gasteiger, 2015) developed a methodology that can routinely be applied to real measurements; it is used in this paper. An alternative model was used by Madonna et al. (2018) and applied to CL51 and CS135 measurements during INTERACT-II.

In summer 2015 a dedicated campaign CeiLinEx2015 ("ceilometer intercomparison experiment") was set up to better un-

derstand the performance of several commercially available ceilometers. In this paper we use data from this campaign to investigate whether signals can successfully be corrected for water vapor absorption. After a brief introduction to CeiLinEx2015 (next section) we discuss several approaches for the validation of the water vapor correction (Section 3). In the key part of our paper we discuss the main features of the validation procedure, especially the selection of the validation range and the spectral extrapolation, and select three representative atmospheric cases to scrutinize the validation. A short summary concludes the

paper.

## 2   CeiLinEx2015: description and objectives

To support E-PROFILE and TOPROF the Meteorological Observatory Hohenpeißenberg of the German Weather Service (DWD) has initiated an intercomparison campaign (CeiLinEx2015) at the Meteorological Observatory Lindenberg of the DWD



**Table 1.** List of deployed ceilometers in CeiLinEx2015: providers are DWD (German Weather Service), LMU (Ludwig-Maximilians-Universität München), RUB (Ruhr-Universität Bochum), GCRI (Global Change Research Institute), and CSci (Campbell Scientific, Manufacturer of the instruments). The emitted wavelength is given in nm, the vertical coverage in km.

| ID | Manufacturer | Type | Owner | Wavelength | Vert. Cov. |
|----|-----|------|-------|-----------|-----------|
| CHM-1 | Lufft | CHM15k | DWD | 1064 | 15.4 |
| CHM-2 | Lufft | CHM15k | DWD | 1064 | 15.4 |
| CHX-1 | Lufft | CHM15kx | DWD | 1064 | 15.4 |
| CHX-2 | Lufft | CHM15kx | LMU | 1064 | 15.4 |
| CL51-1 | Vaisala | CL51 | DWD | $\approx 910$ | 15.4 |
| CL51-2 | Vaisala | CL51 | GCRI | $\approx 910$ | 15.4 |
| CL31-1 | Vaisala | CL31 | DWD | $\approx 910$ | 7.7 |
| CL31-2 | Vaisala | CL31 | RUB | $\approx 910$ | 7.7 |
| CS-1 | Campbell | CS135 | CSci | $\approx 912$ | 7.7 |
| CS-2 | Campbell | CS135 | CSci | $\approx 912$ | 7.7 |
| LD-1 | Vaisala | LD40 | DWD | 855 | 15.3 |
| LD-2 | Vaisala | LD40 | DWD | 855 | 15.3 |

in Lindenberg, Germany (52.209 N, 14.122 E, 120 m above msl). It took place from 1 June till 15 September 2015. Twelve ceilometers were deployed for continuous measurements: all instruments are commercially available systems as they are used by observational networks, service providers, or research institutes. On the one hand instruments from different manufacturers were set up, and different types from the same manufacturer were considered. On the other hand two instruments of each type

were installed to get an rough impression on the "instrument-to-instrument" variability. An overview of the deployed instruments is given in Table 1. The first column lists the acronyms of the instruments as they are used in our investigation. Note, that the last column gives the vertical coverage of the data sets, that is larger than the range of data exploitable in a meteorological sense. The time resolution of "raw" data is in the range of 15–30 seconds and the spatial resolution is 10–15 m. CeiLinEx2015 was the first campaign since the WMO international ceilometer intercomparison (Jones et al., 1988) in 1986 where six different

types of ceilometers from Vaisala, Lufft and Campbell Scientific were compared. According to the manufacturers the emitted wavelength of the CL31 and CL51 is 910 ± 10 nm, and 912 nm for the CS135. As the Campbell ceilometer was temperature-controlled it is expected that this wavelength is quite stable; the spectral bandwidth is ± 3.5 nm. Lufft's CHM15kx is a special version of the standard CHM15k-ceilometer with tilted optical axes and a larger field-of-view to reduce the range of incomplete overlap. Note, that the quite old LD40 ceilometers are not considered in this study.

The main goals of CeiLinEx2015 were twofold: the characterization of instruments and the retrieval of optical properties of aerosols ($\beta_p$). The former comprises the investigation of overlap properties, identification of measurement artifacts, and studies on the instrument's sensitivity to e.g. changes of the ambient temperature. The latter includes the calibration of the systems and the correction of the signals for water vapor absorption. Water vapor absorption is relevant for the Vaisala and Campbell




ceilometers. Moreover, specific topics as the comparison of derived cloud base heights and the derivation of the mixing layer height were covered.

Four radiosondes per day are available in Lindenberg: at 00, 06, 12, and 18 UTC. Profiles of the air density, calculated from the measured temperature and pressure profiles, are used for the Rayleigh calibration. Measurements of the relative humidity

are required for the water vapor correction. Ancillary data also include measurements from an AERONET (Holben et al., 1998) sun photometer, providing e.g. aerosol optical depths between 340 nm and 1640 nm. This information can be used to extrapolate optical properties between different wavelengths.

Finally, the Polly$^{XT}$ lidar (Baars et al., 2016; Engelmann et al., 2016) RALPH was used as reference for the ceilometer measurements; CeiLinEx2015 was the first application of this instrument. It complies with the standard configuration of the

EARLINET's research lidars (Pappalardo et al., 2014). Note, that depolarization measurements were not relevant in the framework of this investigation. RALPH has been moved to Hohenpeißenberg, Germany, after the campaign to become part of EARLINET.

## 3   Concepts of validation

A strict validation of an "aerosol profile" derived from ceilometer measurements after applying a water vapor correction is

not possible because no independent profile at the same wavelength is available. Thus it is necessary to transform profiles between a "water vapor contaminated" wavelength and another wavelength where high quality data not subject to absorption are available. This extrapolation requires assumptions on the wavelength dependence of the optical properties of particles. Moreover, "technical corrections" for incomplete overlap or signal distortions might be required that are different for the ceilometers under review and the reference system. These are reasons for understanding the term "validation" as sort of an

intercomparison and consistency check. Having this in mind we feel that it is nevertheless allowed to henceforward use the term "validation" to better make clear the purpose and motivation of our investigation.

There are several options for the validation of an "aerosol profile". The most obvious strategies are either the comparison of signals $P(z)$, of attenuated backscatter $\beta^*(z)$ or of particle backscatter coefficients $\beta_p(z)$, being $z$ the height (vertically looking systems). These alternatives are discussed in the following.

### 3.1   Concept based on signals

In the case of considering signals $P$ we determine the ratio of the signal $P(\lambda_{\mathrm{off}}, z)$ at a wavelength that is not affected by water vapor absorption (e.g. $\lambda_{\mathrm{off}}$ = 1064 nm) and $P(\lambda_{\mathrm{on}}, z)$ that is affected (e.g. $\lambda_{\mathrm{on}}$ = 910 nm). This results in a height dependent conversion function $\eta(z)$ and allows to extrapolate from one wavelength to the other. The conversion function $\eta(z)$ is defined as

$$\eta(z) = \frac{P(\lambda_{\mathrm{off}}, z)}{P(\lambda_{\mathrm{on}}, z)} \qquad (1)$$



assuming $\lambda_{\mathrm{on}} < \lambda_{\mathrm{off}}$. The signal at the "water vapor wavelength" $\lambda_{\mathrm{on}}$ is

$$P(\lambda_{\mathrm{on}}, z) = C_L \, \frac{\beta(\lambda_{\mathrm{on}}, z)}{z^2} \, T_m^2(\lambda_{\mathrm{on}}, z) \, T_p^2(\lambda_{\mathrm{on}}, z) \, T_{\mathrm{w,eff}}^2(\lambda_{\mathrm{on}}, z) \tag{2}$$

Here, $T_{\mathrm{w,eff}}$ is the effective transmission due to water vapor absorption. As the emitted spectrum of the ceilometers is much broader than the width of individual absorption lines, an effective transmission representative for $\lambda_{\mathrm{on}}$ is calculated following

Wiegner and Gasteiger (2015). In this context the center wavelength $\lambda_0$ of the emitted spectrum and – to a lesser extent – the full width at half maximum (assuming a Gaussian profile) $\Delta\lambda$ of the spectrum are crucial. $T_m$ and $T_p$ are the transmissions due to Rayleigh scattering and particle extinction, respectively, $C_L$ is the lidar constant, and $\beta$ the backscatter coefficient.

At the "offline" wavelength the signal can be described according to

$$P(\lambda_{\mathrm{off}}, z) = C_L \, \frac{\beta(\lambda_{\mathrm{off}}, z)}{z^2} \, T_m^2(\lambda_{\mathrm{off}}, z) \, T_p^2(\lambda_{\mathrm{off}}, z) \tag{3}$$

For the transformation of the signal between $\lambda_{\mathrm{on}}$ and $\lambda_{\mathrm{off}}$ the lidar constant cancels out because we consider the same instrument. This leads to

$$\eta(z) = \frac{\beta(\lambda_{\mathrm{off}}, z)}{\beta(\lambda_{\mathrm{on}}, z)} \left( \frac{T_m(\lambda_{\mathrm{off}}, z)}{T_m(\lambda_{\mathrm{on}}, z)} \right)^2 \left( \frac{T_p(\lambda_{\mathrm{off}}, z)}{T_p(\lambda_{\mathrm{on}}, z)} \right)^2 T_{\mathrm{w,eff}}^{-2}(\lambda_{\mathrm{on}}, z) \tag{4}$$

The backscatter term $B(z)$ – the first on the right hand side of Eq. (4) – is

$$B(z) = \frac{\beta(\lambda_{\mathrm{off}}, z)}{\beta(\lambda_{\mathrm{on}}, z)} = \frac{\beta_m(\lambda_{\mathrm{off}}, z) + \beta_p(\lambda_{\mathrm{off}}, z)}{\beta_m(\lambda_{\mathrm{on}}, z) + \beta_p(\lambda_{\mathrm{on}}, z)} = \frac{\beta_m(\lambda_{\mathrm{off}}, z) + \beta_p(\lambda_{\mathrm{off}}, z)}{L_m \, \beta_m(\lambda_{\mathrm{off}}, z) + L_p(z) \, \beta_p(\lambda_{\mathrm{off}}, z)} \tag{5}$$

The $\beta_p$-profiles are obtained from a reference lidar operating at the absorption-free wavelength $\lambda_{\mathrm{off}}$. In Eq. (5) we have introduced the ratio $L_p$ that is based on the Angström-approach: we find

$$L_p(z) = \frac{\beta_p(\lambda_{\mathrm{on}}, z)}{\beta_p(\lambda_{\mathrm{off}}, z)} = \left( \frac{\lambda_{\mathrm{on}}}{\lambda_{\mathrm{off}}} \right)^{-\kappa(z)} \approx \frac{\tau_p(\lambda_{\mathrm{on}})}{\tau_p(\lambda_{\mathrm{off}})} \tag{6}$$

with $\tau_p$ as the aerosol optical depth and $\kappa$ the Angström exponent. Note, that here we define $\kappa$ in terms of the backscatter coefficient derived from lidar measurements (e.g. 532 nm and 1064 nm). Mostly the Angström exponent is defined by means

of the aerosol optical depth $\tau_p$, e.g. retrieved from AERONET data. In the latter case it is implicitly considered constant with height, otherwise $\kappa$ can be determined as a height-dependent function. Analogously we get from the Rayleigh theory

$$L_m = \frac{\beta_m(\lambda_{\mathrm{on}})}{\beta_m(\lambda_{\mathrm{off}})} = \frac{\alpha_m(\lambda_{\mathrm{on}})}{\alpha_m(\lambda_{\mathrm{off}})} = \left( \frac{\lambda_{\mathrm{on}}}{\lambda_{\mathrm{off}}} \right)^{-4.08} > 1 \tag{7}$$



In case of an aerosol-free atmospheric layer (e.g., the free troposphere) and the above mentioned wavelengths (910 nm and 1064 nm) $B(z)$ approaches $L_m^{-1} = 0.528$, in the case of a layer where $\beta_p \gg \beta_m$ is fulfilled $B(z) \approx L_p^{-1}$, e.g. $B(z) = 0.855$ if $\kappa = 1$.

The second term on the right hand side of Eq. (4) is calculated readily by

$$\left( \frac{T_m(\lambda_{\text{off}}, z)}{T_m(\lambda_{\text{on}}, z)} \right)^2 = \exp \left\{ -2 \int_0^z \alpha_m(\lambda_{\text{off}}, z') \, (1 - L_m) \, dz' \right\} > 1 \tag{8}$$

For the third term on the right hand side of Eq. (4) we get

$$\left( \frac{T_p(\lambda_{\text{off}}, z)}{T_p(\lambda_{\text{on}}, z)} \right)^2 = \exp \left\{ -2 \int_0^z \alpha_p(\lambda_{\text{off}}, z') \, (1 - L_p(z')) \, dz' \right\} \tag{9}$$

which is typically larger than 1. For Eq. (9) profiles of the particle extinction coefficient $\alpha_p$ must be available from the reference lidar. Note, that here we have used the common assumption that $\kappa$ based on backscatter coefficients (Eq. 6) or based on extinction coefficients (Eq. 9) is the same. This implies, that due to the fundamental relationship

$$\frac{\beta_p(\lambda_{\text{on}})}{\beta_p(\lambda_{\text{off}})} = \frac{S_p(\lambda_{\text{off}})}{S_p(\lambda_{\text{on}})} \frac{\alpha_p(\lambda_{\text{on}})}{\alpha_p(\lambda_{\text{off}})} \tag{10}$$

the lidar ratio $S_p$ is the same at $\lambda_{\text{off}}$ and $\lambda_{\text{on}}$. The validity of this assumption can easily be checked by means of the online-tool MOPSMAP (Gasteiger and Wiegner, 2018) if realistic assumptions of the aerosol type or the microphysical properties are available.

In the case of an atmosphere with height independent Ångström exponent $\kappa$, or if height independence must be assumed due to the lack of range resolved data, Eqs. (8) and (9) can be simplified, and Eq. (4) can be written as

$$\eta(z) = \left( \frac{B(z)}{T_{\text{w,eff}}^2(\lambda_{\text{on}}, z)} \right) T_p^{2(1-L_p)}(\lambda_{\text{off}}, z) \, T_m^{2(1-L_m)}(\lambda_{\text{off}}, z) \tag{11}$$

with the transmissions $T_p$ and $T_m$ at wavelength $\lambda_{\text{off}}$. The vertical profile of $\eta$ is primarily governed by the vertical profile of $B(z)$. With Eq. (4) or Eq. (11) the measured signal at $\lambda_{\text{on}}$ can be transferred to $\lambda_{\text{off}}$ by Eq. (1) or vice versa for intercomparison, i.e., calibration of the systems is not required for this type of validation.

## 3.2 Concept based on attenuated backscatter

From the definition of the attenuated backscatter $\beta^*$

$$\beta^*(\lambda, z) = \frac{P \, z^2}{C_L} \tag{12}$$





and Eq. (1) it is directly clear that the ratio of the attenuated backscatter at the two wavelengths is

$$\frac{\beta^*(\lambda_{\mathrm{off}}, z)}{\beta^*(\lambda_{\mathrm{on}}, z)} = \eta(z) \tag{13}$$

Thus, the validation directly follows the mathematical formalism described in Sect. 3.1 because the underlying physical concept of both approaches is identical.

## 3.3 Concept based on particle backscatter coefficients

If the particle backscatter coefficient $\beta_p$ is used for validation, the signals must be inverted. As shown by Wiegner and Gasteiger (2015) the measured Vaisala-signals are first corrected for water vapor absorption by multiplying with $T_{\mathrm{w,eff}}^{-2}(\lambda_{\mathrm{on}})$. Subsequently a standard inversion technique (Klett, 1981; Fernald, 1984) is applied. This leads to $\beta_p(\lambda_{\mathrm{on}})$ and the extrapolation to $\beta_p$ at $\lambda_{\mathrm{off}}$ can be performed by means of the Angström exponent with the same assumptions mentioned above. These profiles can be compared to inversions of measurements of RALPH. In contrast to the previous options, the inversion however requires the knowledge of the lidar ratio and calibrated signals. In the case of the ceilometers this might be an issue as the signal-to-noise ratio in the free troposphere (under aerosol free conditions) is low, and absolute calibration requires specific atmospheric conditions, i.e. long time series of measurements. As the same lidar ratio is used in both retrievals a possible error of $S_p$ however would not influence the validation.

It is clear that this option is more complicated and includes more error sources. Though $\beta_p(z)$ is a direct property of the particles in height $z$, whereas $P(z)$ and $\beta^*(z)$ do not only depend on aerosol properties in height $z$ but also on properties of the atmospheric path below $z$, we do not select this concept in our investigation.

## 4 Validation: discussion and results

Based on the previous discussion we focus on the validation of signals. In principle two alternative approaches are possible: either water vapor affected ceilometer signals near 910 nm ($\lambda_{\mathrm{on}}$) are extrapolated to 1064 nm ($\lambda_{\mathrm{off}}$) and compared to reference signals of RALPH, or one can extrapolate RALPH-signals to the "water vapor domain" and compare them with ceilometer measurements (Vaisala, Campbell). In this paper we decided to extrapolate the signal with the higher quality, i.e. we choose the second option.

The input required for the determination of the conversion function $\eta$ (Eq. 11) is available from CeiLinEx2015: For calculating $B(z)$ we use $\beta_m$ from the Rayleigh-theory with the air density derived from radiosondes, the transmission $T_{\mathrm{w,eff}}(\lambda_{\mathrm{on}})$ due to water vapor is calculated according to Wiegner and Gasteiger (2015) with the water vapor number density derived from the radiosondes as well, $L_p$ is estimated using the Angström exponent $\kappa$ from AERONET or RALPH data, and $\beta_p$ and $\alpha_p$ are derived from the inversion of co-incident RALPH measurements.

After defining our criteria for a successful validation in the following Section we in detail discuss the vertical range that is suitable for validation (Section 4.2), how the spectral extrapolation of aerosol optical properties is provided (Section 4.3), and



**Table 2.** The three validation cases

| Case | Date | Time |
|------|------|------|
| A | 2 July 2015 | $00 - 03$ UTC |
| B | 20 August 2015 | $05 - 08$ UTC |
| C | 14 August 2015 | $00 - 03$ UTC |

the water vapor correction (Section 4.4). Three cases (see Table 2) that cover relevant atmospheric conditions for the validation are discussed in detail in Section 4.5: one case with average water vapor amount $w$ (Case A), a second case with dry conditions (i.e., $T_{\text{w,eff}}$ is large) and low $\tau_p$ (Case B), and a third case with large water vapor content (i.e., $T_{\text{w,eff}}$ is small) and large $\tau_p$ (Case C). Common to all cases is that the aerosol distribution was quite stable and no low clouds were present.

## 4.1 Definition of criteria

According to the previous section we use Eq. (1) to calculate a hypothetical RALPH-signal at a wavelength in the water vapor regime; only integer numbers are considered.

$$P(\lambda_{\text{on}}, z) = \frac{P(\lambda_{\text{off}}, z)}{\eta(z)} := P_{\text{extra}}(\lambda_{\text{on}}, z) \tag{14}$$

The term $P_{\text{extra}}(\lambda_{\text{on}}, z)$ is introduced to make clear that it is not a measurement but a signal extrapolated to $\lambda_{\text{on}}$. For a quantitative assessment of the agreement between $P_{\text{extra}}(\lambda_{\text{on}}, z)$ and the measured ceilometer signal $P_{\text{ceilo}}(z)$ at an actually unknown wavelength in the "water vapor regime", we define the ratio $F$ as

$$F(\lambda_{\text{on}}, z) = c_{\text{norm}} \frac{P_{\text{ceilo}}(z)}{P_{\text{extra}}(\lambda_{\text{on}}, z)} \quad \text{with} \quad c_{\text{norm}} = \left( \frac{1}{N} \sum_{i=1}^{N} \frac{P_{\text{ceilo}}(z_i)}{P_{\text{extra}}(\lambda_{\text{on}}, z_i)} \right)^{-1} \tag{15}$$

The normalization factor $c_{\text{norm}}$ is chosen as the average over the validation range assuming $N$ range bins $z_i$, $i = 1, \ldots, N$. We call the range from $z_1$ to $z_N$ the "validation range". The choice of the lower range $z_1$ is influenced by the overlap characteristics of the involved systems, the upper range by the signal-to-noise ratio and signal artefacts. These issues are discussed in detail in Section 4.2.

In the case of a correct treatment of the water vapor absorption $F(\lambda_{\text{on}}, z)$ should not depend on the height ($dF/dz = 0$); moreover, due to the normalization, $F(\lambda_{\text{on}}, z)$ should be 1. If the decrease of the measured ceilometer signal with height is stronger ("stronger attenuation") than that of the lidar extrapolated to the selected wavelength $\lambda_{\text{on}}$, i.e. $dF/dz$ is negative, then the assumed water vapor absorption at that wavelength is too small in comparison to the actual absorption. Positive $dF/dz$ corresponds to an overestimation of the absorption.

Consequently, we chose the minimum of the absolute value of the slope $dF/dz$ as the criterion for a correct treatment of the water vapor absorption. From this criterion theoretically the central wavelength of the emitted spectrum $\lambda_{\text{on}}$ can be derived



and compared to the ceilometer's specification. In reality this is however not the case for several reasons: the exact emission spectrum of the laser is unknown, and absorption can be similar at different wavelengths. Note, that $\lambda_{on}$ can be different for different ceilometers and time-dependent. Having this in mind typically several wavelengths should exist where the agreement between a ceilometer and extrapolated RALPH measurements is similar.

5    Additionally, the mean deviation of $F$ from unity in the validation range, $\Delta F$, given in % and defined as

$$\Delta F(\lambda_{on}, z) = \frac{100}{N} \sum_{1}^{N} (F'(\lambda_{on}, z) - 1) \qquad \text{with} \quad F' = \begin{cases} F & \text{for } F \geq 1 \\ F^{-1} & \text{for } F < 1 \end{cases} \qquad (16)$$

can be considered as a score. Finally, to strengthen the above described validation an additional check has been applied; it is related to $F$ but maybe more descriptive. The "decrease of the signal" is estimated by fitting a straight line to the measured $P z^2$ (Vaisala or Campbell ceilometers) between $z_1$ and $z_2 = z_1 + \Delta z$, and described by the ratio $s$ at these two ranges

$$10 \quad \frac{P z_1^2}{P z_2^2} = s = \frac{\beta(\lambda_{on}, z_1)}{\beta(\lambda_{on}, z_2)} T_{\Delta,m}^{-2}(\lambda_{on}) T_{\Delta,p}^{-2}(\lambda_{on}) T_{\Delta,w,eff}^{-2}(\lambda_{on}) \qquad (17)$$

It can be compared with values expected from the lidar equation (right hand side of Eq. 17) with $T_\Delta$ being the transmissions of the layer $\Delta z$ in the "absorption spectral range" caused by the different atmospheric constituents ($m$, $p$, $w$ for air molecules, particles, and water vapor, respectively). Typically, we choose $\Delta z$ as the validation range as defined below. In this context it is assumed that within that layer the ratio $\beta_p(z_1)/\beta_p(z_2)$ is wavelength-independent.

## 4.2   The validation range

To find a suitable validation range the investigation of the range of incomplete overlap of the lidar and the ceilometers is essential. It determines especially the lowest suitable range for the validation. Fig. 1 shows the range corrected reference signal (red solid line) and the corresponding signals (dashed) from the four Lufft ceilometers from Case A: CHX-1 (blue), CHX-2 (green), CHM-1 (red) and CHM-2 (black). All measurements concern the same wavelength $\lambda$ = 1064 nm, and are thus directly comparable. They are scaled to match at 0.7 km, and all ceilometer signals have been smoothed over $\pm$ 3 range bins. In contrast to RALPH, the ceilometer data have undergone an overlap correction. It is determined by the manufacturer for each individual Lufft-ceilometer; indeed they vary from one instrument to another. The corrections were introduced to make different ceilometers deployed in a network, especially for that of the German Weather Service, comparable. It should be recognized that with this information it is possible to consider either overlap corrected profiles or profiles without overlap correction. On the basis of the latter it is in principle possible to apply own overlap correction functions determined from horizontal (e.g., Wiegner et al., 2014) or vertical measurements (e.g., Hervo et al., 2016) under homogeneous aerosol distributions.

It can be seen that the agreement of the signals of the CHM-ceilometers (red and black lines) and RALPH is quite good above 0.5 km, even above the mixing layer up to 4 km. However, in the lowermost 500 m large discrepancies occur: the overlap correction for the CHX-ceilometers (note, that they are not part of the German Weather Service network) only show





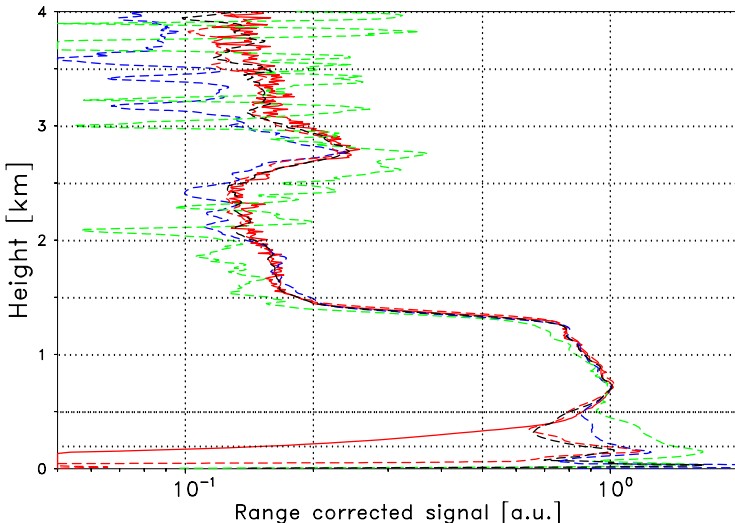

**Figure 1.** Range corrected reference signal (RALPH, red solid line). The dashed lines refer to the Lufft ceilometers: CHX-1 (blue), CHX-2 (green), CHM-1 (red) and CHM-2 (black), all at $\lambda = 1064$ nm and scaled to match at 0.7 km. Measurements concern Case A (see Table 2).

similar shapes whereas the absolute values are quite different. Though the two CHM-ceilometer agree well except in the lowermost range below 80 m they do not agree with the CHX-1 and CHX-2. This underlines the difficulty to determine accurate overlap corrections. Above the mixing layer height the CHX-signals are quite noisy and especially the CHX-2 (green line) shows unrealistic profiles. Investigation of Cases B and C (not shown) in general confirms these conclusions: there is a

good agreement between the two CHM-ceilometers down to approximately 100 m, the overlap correction of the CHX-1 seems to be acceptable but only shows the overall shape, whereas the CHX-2 fails. Again, a surprisingly good agreement between the CHM- and RALPH-measurements in the lowermost 1–2 kilometers of the free troposphere is found.

In Fig. 2 the corresponding intercomparison in the spectral regime of the water vapor absorption is shown. Vaisala ceilometers are shown as dashed lines, Campbell ceilometers as dashed-dotted lines. The reference signal of RALPH (red solid line) has

been extrapolated to (as an example) 910 nm, i.e. water vapor absorption is considered, but the wavelength of the ceilometers is actual unknown. All signals are scaled to match in 0.7 km and smoothed as above. It is immediately clear that the validation range is strongly limited. In this case it is certainly neither below 0.5 km nor above 1.3 km. Inside this range it can be seen that the agreement between the CL51 signals (pink and black dashed lines) and the extrapolated reference signal seems to be almost perfect. In the lowermost part of the troposphere where the signals suffer from incomplete overlap no agreement is found. One

reason, the missing overlap correction for RALPH, has already been mentioned. The two CL51 profiles however do not match either, especially below 0.3 km. This indicates that the generic overlap correction function provided by the manufacturer may





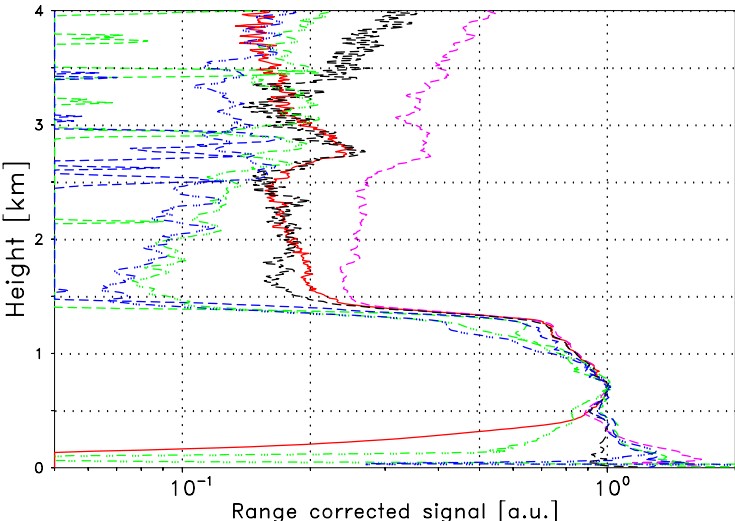

**Figure 2.** Range corrected signals (Case A) of the reference lidar RALPH extrapolated by means of the conversion function $\eta$ to 910 nm (red solid line) and measurements of the ceilometers in the water vapor regime. Vaisala ceilometers are plotted as dashed lines: CL51-1 (pink), CL51-2 (black), CL31-1 (green), and CL31-2 (blue). The dashed-dotted lines correspond to the Campbell ceilometers: CS-1 (green) and CS-2 (blue). All curves are in arbitrary units and scaled to match at 0.7 km altitude.

not be applicable to all CL51 with the same accuracy. The agreement between the two CL31 profiles is quite good, but does not agree with the CL51. No agreement is found between the two CS135 ceilometers, in particular the profile of the CS-1 (green dashed-dotted line) is totally different from the others. This example is in accordance of Fig. 1 and demonstrates that due to the very large uncertainty of the overlap correction a validation of the water vapor correction is impossible in the lowermost
5   atmosphere, where aerosol backscattering is normally the largest.

Comparison of the signals above a height of approximately 1.4 km (Fig. 2) helps to assess the upper range of the validation range. The rapid decrease of the particle backscatter at the transition from the mixing layer to the free troposphere seems to raise problems in the data acquisition of all ceilometers and leads to a quite different drop of the signals. Another issue are signal artefacts characteristic for many ceilometers as described by Kotthaus et al. (2016) for the Vaisala CL31 ceilometer.
10   They also pointed out that a careful check of meta data and the consideration of the firmware version is essential. Obviously the increase of the range corrected signal with height in the free troposphere is in contradiction to realistic signals from an (almost) aerosol free atmosphere (Rayleigh atmosphere). A similar increase but smaller signals are found for the CS135 ceilometers, whereas the signals of the CL31 ceilometers (green and blue dashed lines) are totally attenuated.





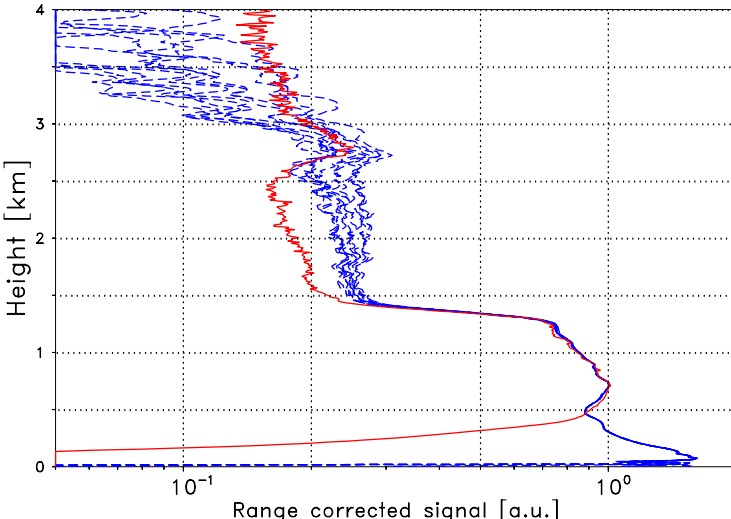

**Figure 3.** Range corrected signals (Case A) of the reference lidar RALPH extrapolated by means of the conversion function $\eta$ to 910 nm (red line) and measurements of the CL51-1 ceilometer corrected by different dark measurements (blue lines). All curves are in arbitrary units and scaled to match at 0.7 km altitude.

From measurements with the "termination hood" – a device that blocks backscattered laser radiation – it is known that mainly the range from 3 km to 8 km is affected by artefacts, with a maximum positive deviation between 4 km and 6 km. These measurements are often referred to as dark measurements. In principle such dark measurements can be used to correct ceilometer signals. The example of Case A shown in Fig. 3 should demonstrate its potential. The blue lines illustrate ten

5    different cases where different dark measurements have been subtracted from the CL51-1 signal. It is clear that on the one hand the slope of the signals in the free troposphere is much more realistic than before (pink dashed line in Fig. 2), on the other hand most of the cases still do not show the slope as expected from Rayleigh scattering (see extrapolated RALPH measurement; red line) and the differences between the 10 profiles are considerable. Indeed dark measurements exhibit a certain temporal variability. Preliminary investigations within CeiLinEx2015 show that there is no significant correlation with temperature, and

10    other reasons have not yet been identified. Accordingly at the present state this kind of correction does not provide the accuracy required to extent the validation range to altitudes above the mixing layer. Further investigations including dark measurements on a regular basis might improve the situation in future.

We conclude that the validation range is limited to the upper part of the mixing layer and has to be individually assessed for each specific measurement period. However, for a given time period the same validation range is used for all ceilometers.





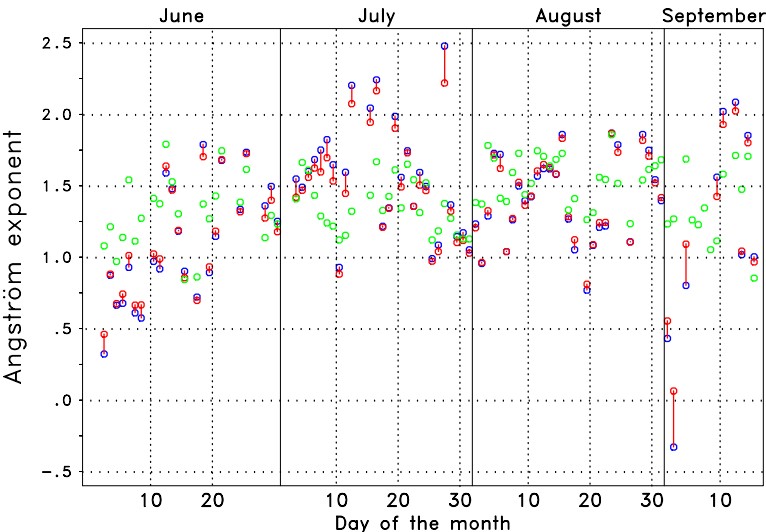

**Figure 4.** Angström exponent (daily averages) for the spectral range 1640/1020 nm (blue, $\kappa_n$) and 1640/870 nm (red, $\kappa_w$), connected by a vertical line. For comparison the standard AERONET output for 870/440 nm (green, $\kappa_a$) is shown.

## 4.3 The spectral extrapolation

For the spectral extrapolation different options based on the Angström exponent are available. The most obvious approach is the use of AERONET data. This data set is well established and it is generally accepted that the accuracy is the best available. Several wavelengths are available so that the range of extrapolation is well covered. The disadvantage of AERONET measure-

5 ments is the limitation to daytime conditions, and the lack of range resolved information as it relies on the aerosol optical depth. Range resolved $\kappa(z)$ can only be derived from a reference lidar system using however a smaller set of wavelengths compared to a sunphotometer. In case of RALPH either an Angström exponent based on backscatter coefficients $\beta_p$ can be determined using measurements at 532 nm and 1064 nm, or an Angström exponent based on extinction coefficients $\alpha_p$ using the Raman channels at 355 nm and 532 nm. In the latter case it is however questionable whether this spectral range is representative for

the wavelength interval from $\lambda_{\mathrm{on}}$ to $\lambda_{\mathrm{off}}$ as $\kappa$ often is wavelength-dependent (e.g., Kaskaoutis and Kambezidis, 2006; Schuster et al., 2006).

AERONET data are available from 27 June until 15 September 2015. As cloudfree conditions are required the temporal sampling is quite inhomogeneous. The measurements at Lindenberg comprises aerosol optical depth (level 2.0 data) at eight wavelengths between 340 nm and 1640 nm. We calculate Angström exponents for three different spectral intervals: the standard

AERONET output for 440/870 nm ($\kappa_a$), and two intervals relevant for the interpolation from $\lambda_{\mathrm{off}}$ to $\lambda_{\mathrm{on}}$: a "narrow" interval



1020/1640 nm ($\kappa_n$) and "wide" interval 870/1640 nm ($\kappa_w$). For the validation we may consider 1-hour, 3-hours and 6-hours averages as well as daily averages, depending on their availability. Note, that a time lag of several hours between the AERONET data and the ceilometer data may occur if the validation period relies on ceilometer measurements during night time. An overview over the three Angström exponents ($\kappa_a$, $\kappa_n$, $\kappa_w$) based on daily averages is shown in Fig. 4. The two Angström

exponents including 1640 nm (red and blue circles) are connected by a red vertical line to facilitate the discrimination from the standard Angström exponent.

The medians of daily averages of $\kappa$ are $\kappa_a = 1.38$, $\kappa_n = 1.42$, and $\kappa_w = 1.37$. For 1-hour, 3-hours and 6-hours averages similar values are found. In total, all values are in the range expected for a continental site as Lindenberg. Low values of $\kappa_w < 0.5$ were observed during two days only, whereas Angström exponents larger than 2.0 were slightly more frequent. On the basis of

individual observations, $\kappa_a$ can be larger or smaller than the NIR-values ($\kappa_n$, $\kappa_w$), and differences larger than 0.5 can occur. This underlines that $\kappa$ can be wavelength-dependent. Due to the high temporal variability shown in Fig. 4 it is recommended to use the Angström exponent closest to the actual ceilometer observations instead of long term averages.

For the validation procedure one shall use $\kappa_w$ in Eq. (6) as only this value completely covers the extrapolation range. To facilitate the reading we omit the subscript $w$ from now on. To estimate the corresponding variability of $L_p$ we again refer to

Fig. 4: applying the median of $\kappa = 1.37$ we get $L_p = 1.239$, whereas for the 10. percentile of $\kappa$ (= 0.81) we get $L_p = 1.135$, and $L_p = 1.331$ for the 90. percentile ($\kappa = 1.83$). This uncertainty together with the relative contribution of particles to the backscatter coefficient at a specific height determine the uncertainty of $B(z)$.

The influence of $\kappa$ on the conversion function $\eta$ is illustrated in Fig. 5, Case A is selected as an example. Three representative wavelengths are displayed with the colors indicating $\lambda_{on} = 905$ nm (red), 910 nm (green) and 915 nm (blue). The full lines

correspond to $\kappa = 1.18$, the dashed to $\kappa = 1.42$ – these values cover the maximum possible range of Angström exponents for Case A (discussed below). The three lines being quite close to each other correspond to three different lidar ratios (45 sr, 55 sr, 65 sr) with $S_p = 45$ sr marked by a circle. In general the profiles of $\eta$ are governed by the height-dependence of $B(z)$: below 0.5 km it is assumed that $\beta_p(z)$ takes the value of $\beta_p$ at 0.5 km. This is a common procedure if an inversion of the lidar data is not possible due to the incomplete overlap. Till the upper part of the mixing layer $\eta$ is dominated by the increasing contribution

of particles whereas above the mixing layer $\eta(z)$ shows a pronounced decrease because $B(z)$ approaches its minimum value in the virtually aerosol-free layers as discussed previously in the context of Eq. (5). It can be seen that $\eta$ strongly depends on $\lambda$ and to a similar or smaller extent on $\kappa$ whereas the dependence on $S_p$ is virtually negligible. As a consequence we use $S_p = 55$ sr for all validations of the ceilometer signals discussed below.

If the microphysical properties of particles significantly change with height, e.g. due to different aerosol types or due to

strong hygroscopic growth, $\kappa$ will become height-dependent. Then, for the assessment of $\kappa$ the availability of $\beta_p$-profiles (see Eq. 6) derived from measurements of a (at least) dual-wavelength lidar is mandatory. In the case of most aerosol lidars the suitable wavelengths are 532 nm and 1064 nm, an interval that unfortunately is quite wide compared to the differences of $\lambda_{on}$ and $\lambda_{off}$. To estimate the relevance of this effect we again consider Case A and assume two cases of an idealized height-dependence: an increase from 90 % to 110 % of a given Angström exponent between the surface and the upper boundary of

the mixing layer (here 1.3 km), and the corresponding decrease. In Fig. 6 the conversion function $\eta$ for the same wavelengths





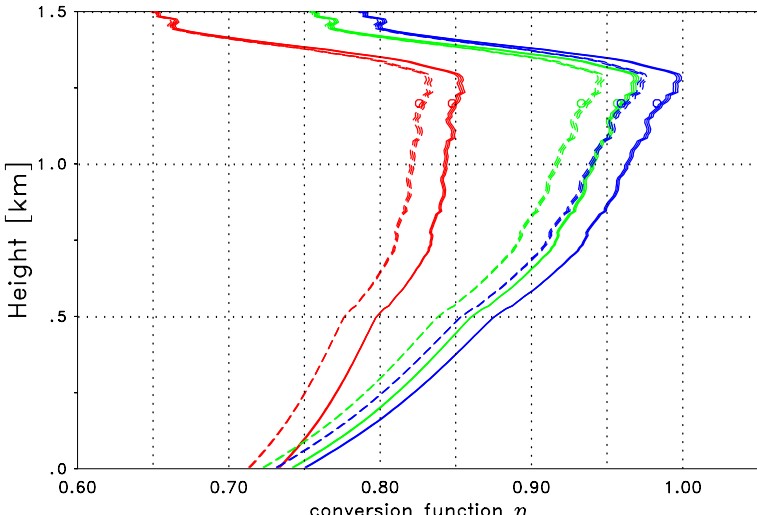

**Figure 5.** Conversion function $\eta$ at 905 nm (red), 910 nm (green) and 915 nm (blue) for Case A, $\kappa$ is assumed to be constant with height. The solid lines are for $\kappa = 1.18$, the dashed lines for $\kappa = 1.42$. The three lines grouping together refer to different lidar ratios with the smallest ($S_p = 45$ sr) marked with a circle.

as before are shown (indicated by the colors) and two mean Angström exponents with $\kappa = 1.18$ and $\kappa = 1.42$ as solid and dashed lines, respectively. The cases with an increasing or decreasing $\kappa$ are marked with crosses and circles, respectively. The remaining profile is based on a constant $\kappa$, already shown in Fig. 5. As mentioned above, $S_p = 55$ sr is assumed. Fig. 6 reveals that a height-dependence of $\kappa$ can have an influence on $\eta$ larger than the influence of $S_p$. Though a generally valid magnitude cannot be assessed because of the variability of $\eta$ with the atmospheric conditions (e.g. water vapor and aerosol distribution) and the spectrum of the laser source, this example demonstrates that the height-dependence of $\kappa$ should be considered whenever reliable data are available. The difference of $\eta$ between height-dependent and height-independent Angström exponents itself is height-dependent. A detailed discussion of different treatments of the spectral dependence is provided for each case study in Section 4.5.

## 4.4  The water vapor profiles

The profile of the water vapor concentration is required to determine $T_{\text{w,eff}}$. It can be readily calculated as described in Wiegner and Gasteiger (2015). A good indication for the overall influence of the water vapor correction on the validation is the total water content per unit area $w$ (in kg/m$^2$, "precipitable water") as it determines the minimum transmission. Typically $T_{\text{w,eff}}$ is virtually constant above 5 or 6 km due to the very low water vapor content above these heights. The relation between $w$ and



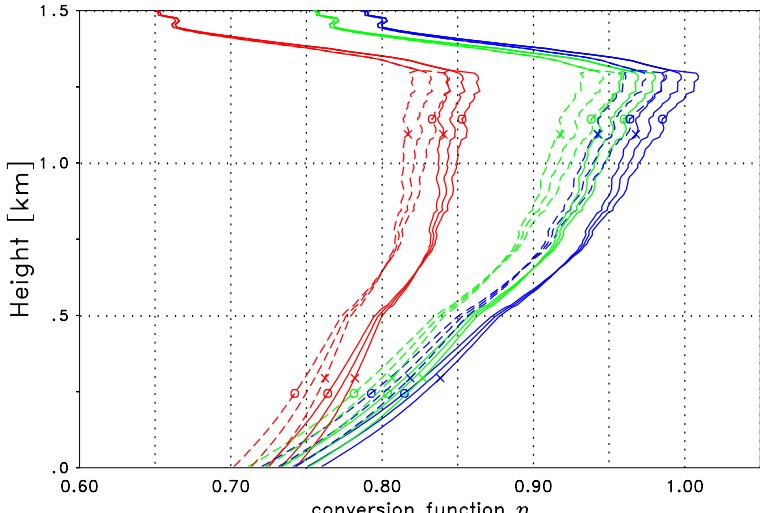

**Figure 6.** Conversion function $\eta$ at 905 nm (red), 910 nm (green) and 915 nm (blue) for Case A. Same as Fig. 5 but with idealized height-dependent $\kappa$. The solid lines are for a mean Ångström exponent of $\kappa = 1.18$, the dashed lines for a mean $\kappa = 1.42$. The three lines grouping together refer to different $\kappa$-profiles: $\kappa$ decreasing and increasing with height is marked with a circle and cross, respectively, with the remaining curve showing the constant $\kappa$ (height independent); see text for details

$T_{\mathrm{w,eff}}$ for $z$=10 km, henceforward referred to as $T_{\mathrm{w,eff}}^{\min}$, for three wavelength $\lambda_{\mathrm{on}}$ (905, 910, 915 nm) is shown in Fig. 7. For example, $T_{\mathrm{w,eff}}^{\min}$ at 910 nm (green dots) is approximately 0.856 and 0.730 for a vapor content of $w = 12$ kg/m$^2$ and $w = 40$ kg/m$^2$, respectively. Between $w = 20$ kg/m$^2$ and $w = 30$ kg/m$^2$, the transmission changes by $dT_{\mathrm{w,eff}}^{\min}/dw \approx 0.0043$ m$^2$/kg. At $\lambda_{\mathrm{on}} =$ 905 nm (red dots) the water vapor absorption is weaker and the sensitivity smaller (0.0029 m$^2$/kg), at 915 nm (blue dots) the

5    opposite is true (0.0049 m$^2$/kg). The small "scattering" of the dots around a perfect line is caused by the fact that different water vapor profiles can result in the same $w$. The range of the actual total water vapor content between 27 June until 15 September $w$ is shown in Fig. 8. This overview helps to select interesting conditions for the case studies discussed in Sect. 4.5. The median of the water vapor content is $w = 21.6$ kg/m$^2$ (average $w = 23.2$ kg/m$^2$), with the 10. percentile and the 90. percentile being $w = 14.1$ kg/m$^2$ and $w = 34.7$ kg/m$^2$, respectively (blue lines). Together with Fig. 7 we can directly estimate the magnitude of the

10    "water vapor correction". If it is compared to the transmission of the air molecules $T_m$ at 1064 nm (not shown) it is obvious that the water vapor effect is much more relevant. If we consider the profile of the median and the percentiles (10., 90.) of $T_m$ of all radiosonde ascents during CeiLinEx2015 we find that $T_m > 0.995$ throughout the troposphere and that the variability – expressed as the difference between the two percentiles – is smaller than $1.4 \cdot 10^{-4}$, i.e., virtually negligible.





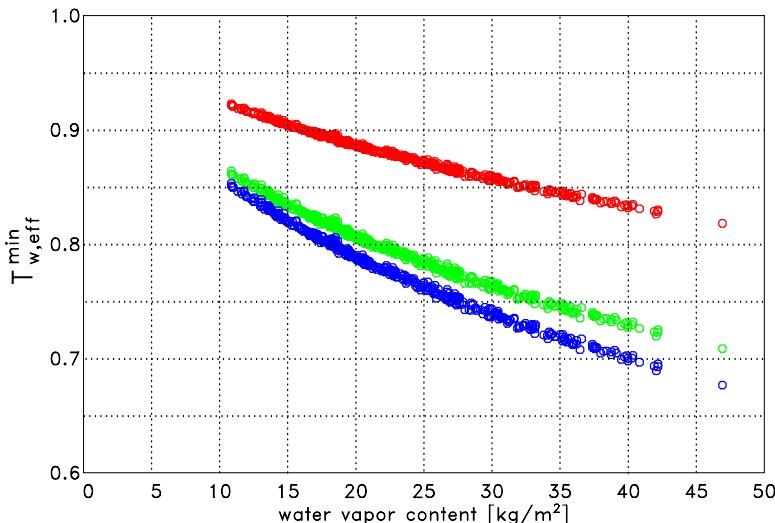

**Figure 7.** Relation between the total water vapor content $w$ and the water vapor transmission at 10 km, $T_{\text{w,eff}}^{\text{min}}$, determined from all radiosonde ascents between 27 June and 15 September 2015. The central wavelength of the laser spectrum is set to 905 nm (red), 910 nm (green) or 915 nm (blue).

We conclude that in the framework of the validation we may use the same profile of the "Rayleigh transmission" whereas individual measurements shall be used for the water vapor profile and the spectral dependence of the aerosol extinction.

### 4.5 Results: The water vapor correction

#### 4.5.1 Case A: 2 July 2015

5 The first case study concerns a typical case with respect to the water vapor abundance. Measurements are taken from 2 July 2015. An overview of the aerosol distribution is shown in Fig. 9 as a time-height cross section of the range corrected signal of the CHX-2 ceilometer (in arbitrary units, logarithmic scale). For the sake of clarity, only 12 hours are shown and the maximum height is limited to 7 km though the maximum range of the ceilometer is 15.4 km. It can be seen that until noon aerosol particles were mainly confined to the lowermost 1.5 kilometers. In the free troposphere aerosol free conditions seem to occur.

10 Until 07 UTC an elevated residual layer is visible, then convection drives the build-up of the mixing layer with a maximum depth of 1.7 km. From a lidar perspective such a fair weather situation is considered as "quite stable". For the validation we select RALPH- and ceilometer-measurements averaged from 00 UTC to 03 UTC to avoid daylight. Based on the criteria described in Section 4.2 the validation range is set to $0.7 < z < 1.3$ km.





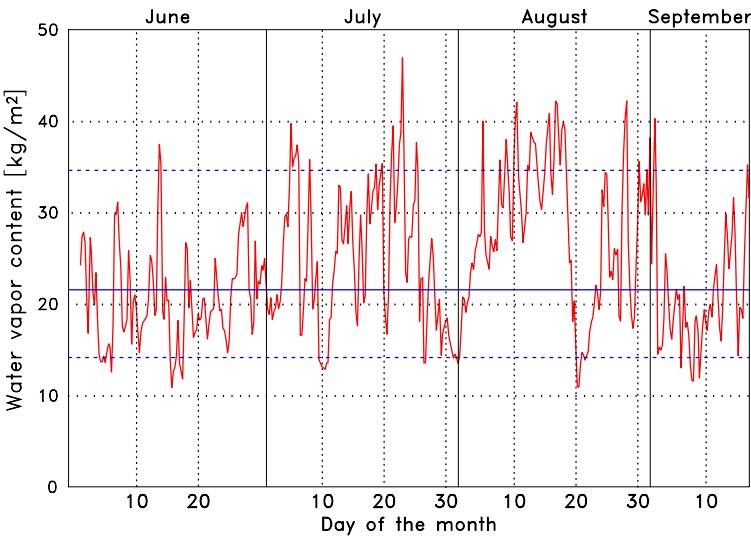

**Figure 8.** Total water vapor content ("precipitable water") $w$ in kg/m$^2$ for each radiosonde ascent during the CeiLinEx2015 campaign. The horizontal lines indicate the median (21.6 kg/m$^2$, solid) and the 10. percentile (14.1 kg/m$^2$) and 90. percentile (34.7 kg/m$^2$, dashed)

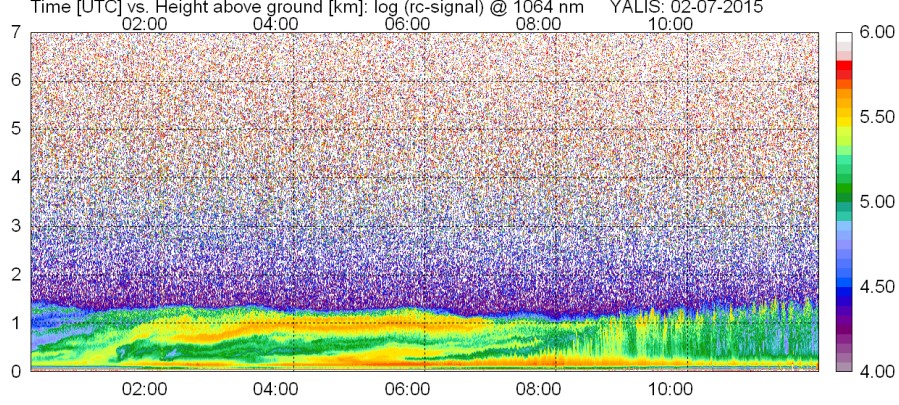

**Figure 9.** Time-height cross section of the range corrected signal (in arbitrary units, logarithmic scale) of CHX-2 from 2 July 2015 (including Case A) until noon. Time is given in UTC, and the height above ground in km; note, that the maximum height shown is not the full measurement range of the ceilometer.



**Table 3.** Effective water vapor transmission at 10 km height, $T_{\mathrm{w,eff}}^{\mathrm{min}}$, for different central wavelengths $\lambda_{\mathrm{on}}$ (in nm) of the laser emission spectrum (water vapor profile of 2 July 2015).

| $\lambda_{\mathrm{on}}$ | 900 | 902 | 904 | 905 | 907 | 910 | 912 | 915 | 918 | 920 | 922 | 925 |
|---|---|---|---|---|---|---|---|---|---|---|---|---|
| $T_{\mathrm{w,eff}}^{\mathrm{min}}$ | 0.811 | 0.838 | 0.888 | 0.894 | 0.850 | 0.817 | 0.818 | 0.801 | 0.831 | 0.883 | 0.897 | 0.877 |

The determination of the Ångström exponent $\kappa$ was complicated as no Level 2.0 data were available for 2 July; gaps of a few days in the AERONET record occur occasionally. If the closest daily average before (30 June, $\kappa = 1.18$) and after the measurements (3 July, $\kappa = 1.42$) are considered quite large temporal differences have to be accepted, reducing the credibility of the values. For this reason we rather rely on Level 1.0 data; here AERONET measurements from the morning of 2 July were available and a mean out of 21 measurements with $\kappa = 1.18 \pm 0.03$ was found. For the validation this corresponds to $L_p \approx$ 1.203 ±0.006.

The scheduled 00 UTC-radiosonde was launched at 22:50 UTC of the day before and provided the profiles required for the water vapor correction. Fig. 10a shows the water vapor profile in terms of the relative humidity (black line, upper scale in percent), and the water vapor number density $n_w$ (red lines, lower scale in $10^{24}$ molecules/m$^3$). For comparison and as indication of the temporal variability the number density from the subsequent radiosonde ascent (6 hours later) is shown as well (dashed line). The water content was 18.3 kg/m$^2$, thus, slightly lower than the median. In the validation range (yellow area) the relative humidity increases with height from 35% to 65%, i.e. it stays in a range where hygroscopic growth of hydrophilic aerosols (if present) is typically moderate.

The particle transmission $T_p$ at 1064 nm is calculated from the RALPH measurements applying the backward Klett algorithm. We use a lidar ratio of $S_p = 55$ sr at 1064 nm, and assume an uncertainty of $\pm 10$ sr for the $\alpha_p$-retrieval. The reference height for the Rayleigh calibration is set to 5.47 km. Because of the incomplete overlap of the lidar we assume that the particle extinction coefficient at 0.5 km does not change below that height. Above the reference height a constant $T_p$ is assumed, i.e., we suppose aerosol free conditions. The resulting profiles of $T_p$ for three lidar ratios are shown in Fig. 10b: it can be seen that for $S_p = 55$ sr (green line) the transmission is $T_p > 0.97$ for all heights. A lidar ratio of $S_p = 45$ sr (red) and $S_p = 65$ sr (blue) lead to a quite small change in $T_p$, increasing with height but never exceeding 0.5%. The same is true for $T_p^{2(1-L_p)}$ which appears in Eq. (11). The $\beta_p(z)$-profile from the same Klett inversion is used to calculate $B(z)$. For reasons of consistency this implies that the backscatter coefficient $\beta_p$ is assumed to be constant in the lowermost 0.5 km.

The effective water vapor transmission $T_{\mathrm{w,eff}}$ is shown in Fig. 10c: the different lines refer to different wavelengths $\lambda_{\mathrm{on}}$ between 900 nm and 925 nm; the width of all spectra is set to 3.5 nm. For example the transmission at 5 km decreases from 905 nm, 925 nm, 907 nm, 910 nm, 900 nm to 915 nm. The minimum transmission $T_{\mathrm{w,eff}}^{\mathrm{min}}$ for a broader range of wavelengths is summarized in Table 3. It can be seen that the minimum transmission varies between $0.8 < T_{\mathrm{w,eff}}^{\mathrm{min}} < 0.9$ depending on the wavelength: minimum absorption occurs between $904 \leq \lambda_{\mathrm{on}} \leq 905$ nm and $920 \leq \lambda_{\mathrm{on}} \leq 924$ nm, whereas absorption is strongest between $913 \leq \lambda_{\mathrm{on}} \leq 916$ nm. As a consequence different wavelengths may result in virtually the same transmission.





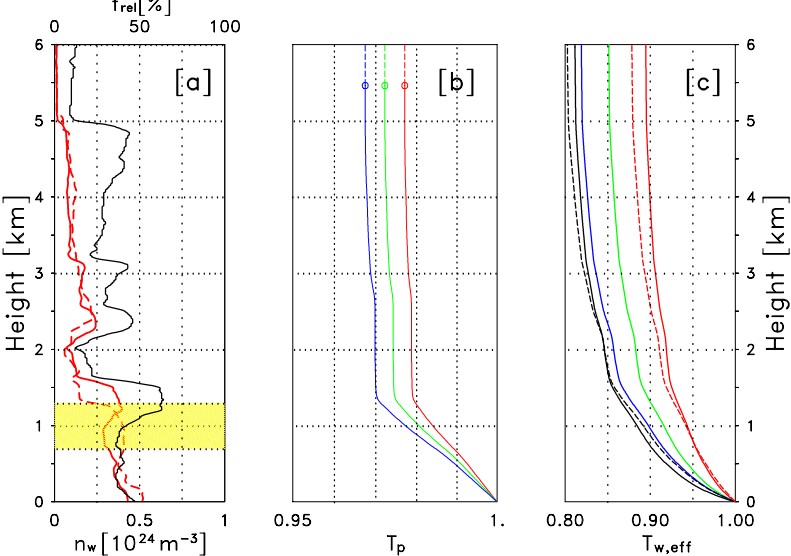

**Figure 10.** (a) Profile of the relative humidity (black line) in percent, see labels at the top, and profiles of the water vapor number density (in $10^{24}$ molecules/m$^3$, labels at the bottom) for the 00 UTC ascent (solid red line) and the 06 UTC ascent (dashed red line). The validation range is indicated by the yellow area. (b): Particle transmission $T_p$ at 1064 nm derived the from Klett inversion of RALPH signals (2 July 2015, 00 UTC – 03 UTC, Case A) assuming a lidar ratio of 45 sr (red), 55 sr (green), and 65 sr (blue), respectively. The circles indicate the reference height. (c): Effective water vapor transmission $T_{w,eff}$ for different laser wavelengths $\lambda_{on}$ (solid lines: 900 nm (black), 905 nm (red), 907 nm (green), 910 nm (blue); dashed lines: 915 nm (black), 925 nm (red)).

When compared to Fig. 7 it is obvious that – especially in the range around 907 nm and 918 nm – the transmission is much more sensitive to errors of the assumed wavelength $\lambda_{on}$ than to errors of the water vapor content. It can reach values of about $dT_{w,eff}^{min}/d\lambda > 0.02\ \mathrm{nm}^{-1}$. In this context it is relevant that in the case of Vaisala ceilometers the emitted spectrum is temperature dependent. A quantitative assessment of this dependence is however not yet available.

5    With this input the conversion function $\eta$ can be determined. Examples for two representative wavelengths are displayed in Fig. 11, with the colors indicating $\lambda_{on}$ = 905 nm (red) and 915 nm (blue). According to Fig. 10c the effective water vapor transmission is largest at 905 nm, and thus $\eta$ takes the smallest values (Eq. 11). The dashed lines show the conversion function if a constant $\kappa$ is assumed: the short-dashed line corresponds to the smallest value of the assumed $\kappa$-range, the long-dashed to the largest value. The lidar ratio is set to $S_p$ = 55 sr. Note, that only the values within the validation range are relevant (yellow

10   background), below that range e.g. the incomplete overlap alters the values. The full lines are derived if a height dependent Angström exponent as derived from the particle backscatter coefficients at 532 nm and 1064 nm is used. The Angström exponent shows an almost linear increase from $\kappa$ = 1.04 to $\kappa$ = 1.21 within the validation range (not shown). This suggests





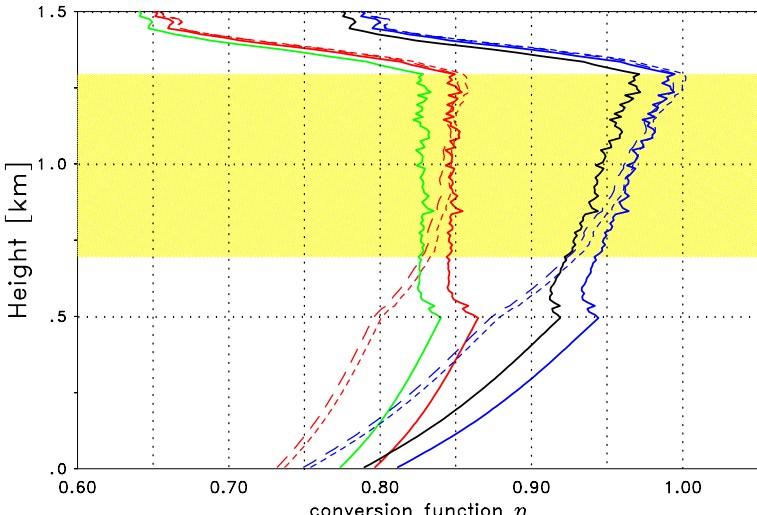

**Figure 11.** Conversion function $\eta$ at 905 nm (red) and 915 nm (blue) for Case A: the short- and long-dashed lines are for constant $\kappa$ with the lower and upper range of of 1.15 and 1.21, respectively; $S_p$ = 55 sr is assumed. The solid lines are calculated with a height dependent $\kappa$ as derived from inverted $\beta_p$-profiles of RALPH measurements. The green and black lines are for 905 nm and 915 nm, respectively, but with a different boundary value used in the inversion. The validation range between 0.7 km and 1.3 km is highlighted in yellow (see text for details).

decreasing particle size, thus hygroscopic growth seems to be not dominant here. Note, that the retrieved $\kappa$-values match very well with the mean Angström exponent from the AERONET data ($\kappa$ = 1.18, see above). Consequently the solid (red or blue) line lies between the corresponding dashed lines in Fig. 11 in the upper part of the validation range. The uncertainty of the height-dependent $\kappa$ is slightly influenced by the sensitivity of $\beta_p$ at 532 nm on the lidar ratio, i.e. $\pm$ 0.5 % and $\pm$ 3 % for the

5    lower and upper boundary of the validation range.

      To extend the discussion we briefly consider the uncertainty that may be caused by the uncertainty of the Rayleigh calibration height. The 1064 nm signal of RALPH suggests that heights around 2.4 km and 5.6 km are suitable as calibration height, however, the signal at 532 nm has a small offset above 4 km. Consequently, $\kappa$ determined from $\beta_p$-retrievals calibrated at the upper calibration height can be used to investigate a worst case scenario. The resulting Angström exponent is considerable larger ($1.22 < \kappa < 1.44$), but again a linear increase with height is found. To illustrate this effect the corresponding conversion

10   functions $\eta$ are shown as green and black solid lines for 905 nm and 915 nm, respectively. They are therefore shifted to smaller values but the vertical dependence is virtually unchanged, compare e.g. the red and the green line in Fig. 11. In most cases,





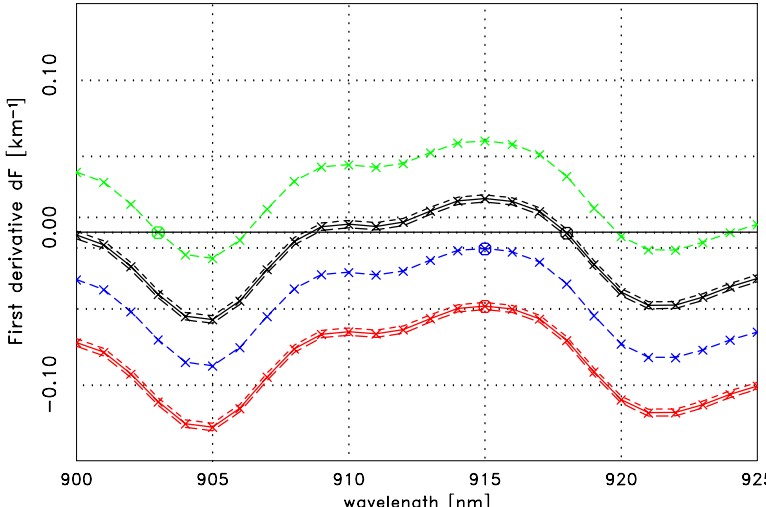

**Figure 12.** First derivative $dF(z, \lambda_\mathrm{on})/dz$, see Eq. (15), of the ratio of the CL51 ceilometer signal and the extrapolated reference lidar signal as a function of $\lambda_\mathrm{on}$: CL51-1 (solid black line) and CL51-2 (solid red line) assuming a height-dependent $\kappa$. The short-dashed and long-dashed lines are for the minimum and maximum $S_p$-values, respectively. The integer wavelength corresponding to the minimum of the absolute values of $dF/dz$ is indicated by a circle. The green (CL51-1) and blue (CL51-2) dashed lines corresponds to $dF/dz$ assuming a constant $\kappa$ as derived from AERONET for comparison. All curves concern Case A.

e.g. if backscatter signals at more than one wavelength are available, retrievals based on an incorrect Rayleigh calibration can however be recognized and thus avoided.

The results of the validation in terms of $dF/dz$ as a function of wavelength are shown in Fig. 12. For an extensive discussion the two options introduced above are considered again: the assumption of a constant $\kappa$ from AERONET and a height-dependent

5   $\kappa$ from the RALPH-data inversion. The solid black line corresponds to the CL51-1, the red line to the CL51-2 measurements assuming a height-dependent $\kappa$ and the default lidar ratio of 55 sr. The short-dashed and long-dashed lines correspond to $S_p$ = 45 sr and $S_p$ = 65 sr, respectively, to demonstrate the quite small uncertainty associated with the uncertainty of the lidar ratio applied in the inversion of the RALPH-data. For comparison, $dF/dz$ assuming a constant $\kappa = 1.18$ is shown as dashed green (CL51-1) and blue (CL51-2) lines.

10   When considering the height dependent $\kappa$ we find the best agreement in the case of the CL51-1 measurements at $\lambda_\mathrm{on}$ = 918 nm with a slope $dF/dz = -3.9 \cdot 10^{-4}$ km$^{-1}$ (marked by a circle in Fig. 12). The mean deviation $\Delta F = 0.9$ % is quite small. The wavelength $\lambda_\mathrm{on}$ = 918 nm is one of the wavelengths where water vapor absorption is comparably weak (cf. Table 3). Accordingly and obvious from Fig. 12, similar absolute values of the slope (and $\Delta F$) are found when the reference signal is





extrapolated to the wavelength of 900 nm or to a wavelength between $908 \leq \lambda_{\mathrm{on}} \leq 912$ nm – the "quality of the agreement"
is virtually indistinguishable. The very small values of $dF/dz$ suggest a perfect water vapor correction, especially when non-integer values are considered as well. In the case of CL51-2 the best agreement is found for $\lambda_{\mathrm{on}}$ = 915 nm with $dF/dz$ =
$-4.8 \cdot 10^{-2}$ km$^{-1}$ (red circle in Fig. 12). Similar values are found in the range 914–917 nm, pointing at the strong part of

the water vapor absorption band. The slope of the ratio is however almost two orders of magnitude larger than in the case of
CL51-1, but still suggests a reasonable water vapor correction. The mean deviation $\Delta F$ = 1.2 % is somewhat larger compared
to the CL51-1 evaluation.

In the case that the constant $\kappa$ from AERONET is used in the water vapor correction the conclusions are similar for the
CL51-1. Inspection of the green curve (Fig. 12) shows, that again wavelengths can be found where the water vapor correction

is perfect, e.g. 902 nm, 906 nm, 920nm, or 924 nm. The best agreement is found for 902 nm. The fact that this is a different
wavelength than in the case of a height-dependent $\kappa$ is irrelevant as long as the spectral emission of the laser is unknown.
The minimum values of $dF/dz$ in the case of CL51-2 (blue curve) are also very small underlining a very good water vapor
correction. Somewhat surprising is that for Case A the constant $\kappa$ leads to better results than the height-dependent $\kappa$. This
might be an effect of the long averaging time and the specific meteorological conditions.

We want to emphasize that this procedure does not allow to retrieve the central wavelength of the laser spectrum. Reasons
are not only the spectral ambiguity of the effective absorption as shown in Fig. 12, but also a certain degree of freedom in
the choice of the validation range, and how to weight the agreement in different altitudes. Nevertheless, the intercomparison
demonstrates that a wavelength in the likely range of the laser emission can be found that leads to a very good agreement of the
signals, in particular in the case of CL51-1. To emphasize this statement the ratio of the measured ceilometer signal (CL51-1

and CL51-2, respectively) and the original lidar signal at 1064 nm has been calculated: they show significantly larger slopes
with $dF/dz$ = $-0.06$ km$^{-1}$ and $-0.13$ km$^{-1}$, respectively. Such negative values are consistent with the fact that water vapor
does not absorb at 1064 nm. This example confirms that the water vapor correction indeed improves the aerosol retrieval.

To underline the correctness of signal slopes discussed above we have calculated the decrease of the signals $s$, see Eq. (17),
in the validation range with $z_1$ = 0.7 km to $z_2$ = 1.3 km. The ratio of the backscatter coefficients is $1.27 \pm 0.01$. The contribution

of the particles is calculated according to the Klett-inversion of the RALPH signals. We assume the same aerosol type within
the layer, thus the ratio $\beta_p(z_1)/\beta_p(z_2)$ is wavelength independent and can be used for $\lambda_{\mathrm{on}} \approx 910$ nm as well. The Rayleigh
contribution to $\beta$ is calculated as usual from the air density derived from the radiosonde data. The transmission of the layer due
to Rayleigh scattering $T_{\Delta,m}$ is virtually 1, and due to particle extinction $T_{\Delta,p} = 0.986 \pm 0.002$ depending on the lidar ratio as
discussed above (Fig. 10b). This is equivalent to $T_{\Delta,p}^{-2} = 1.029 \pm 0.005$ used in Eq. (17). The effective water vapor transmission

of the layer is between $T_{\Delta,\mathrm{w,eff}} = 0.977$ at $\lambda_{\mathrm{on}} = 905$ nm as the lowest effective absorption, and $T_{\Delta,\mathrm{w,eff}} = 0.955$ at $\lambda_{\mathrm{on}} =$
915 nm (strongest absorption, see Fig. 10c). So the last term on the right hand side of Eq. (17) should be between 1.047 and
1.096. From these estimates $s$ should be in the range $1.35 < s < 1.45$. Actually, we find $s = 1.44$ and $s = 1.50$ for CL51-1 and
CL51-2, respectively, which is reasonably close to this range and confirms the better water vapor correction of the CL51-1.

The same kind of validation is attempted for the other ceilometers. An overview together the values given above is summa-

35  rized in Table 4.



**Table 4.** Key parameters of the validation for Case A. The minimum slope $dF/dz$ for an integer wavelength is given, or $dF/dz = 0$ if the corresponding curve shown in Fig. 12 crosses the zero-line (for an non-integer wavelength). According to Eq. (17) the decrease of the range corrected signal $s$ should be $1.35 < s < 1.45$

| Ceilometer | $dF/dz$ | $s$ |
|---|---|---|
| CL51-1 | 0 | 1.44 |
| CL51-2 | $-4.8$E-2 | 1.50 |
| CL31-1 | $-3.0$E-1 | 1.76 |
| CL31-2 | $-1.9$E-1 | 1.63 |
| CS-1 | $-8.9$E-1 | 2.56 |
| CS-2 | $-1.1$E-0 | 2.95 |

For both CL31-ceilometers the decrease of the signals in the validation range was calculated according to Eq. (17), an illustration is already available in Fig. 2. For the CL31-1 (green dashed line) and CL31-2 (blue dashed) we find $s = 1.76$ and $s = 1.63$, respectively, and absolute values of the slope $dF/dz$ that are much larger than in case of the CL51. Such a strong decrease cannot be explained by water vapor absorption at wavelengths around 910 nm. As a consequence we assume that the

reason for the decrease of the signals is the low pulse energy of the CL31 compared to the CL51 ceilometers (1.2 $\mu$J vs. 3 $\mu$J). This hypothesis is supported by the fact that immediately above the top of the mixing layer (approximately at 1.35 km) the signals of both CL31 are totally attenuated. The profiles of both CS135 ceilometers are also shown in Fig. 2 (dashed-dotted lines). It is obvious that the slope of the range corrected signal in the upper part of the mixing layer is much larger than in the case of all Vaisala ceilometers and the reference signal: in the validation range a decrease by a factor $s = 2.56$ (CS-1, blue line)

and $s = 2.95$ (CS-2, green line) and very large negative slopes (see Table 4) are observed that is far beyond what can be caused by water vapor absorption according to Eq. (17). So again we conclude that the shape of the signals is dominated by currently unknown problems. The wavelength of the CS135 is however relatively stable due to the temperature control of the laser so a wavelength drift is unlikely to be an issue. It might be possible that a further reduction of the validation range would help, however, a vertical extent of 0.6 km is already relatively small.

### 4.5.2   Case B: 20 August 2015

As a second case study we selected the period from 05 UTC to 08 UTC of 20 August 2015, referred to as Case B, with quite low total water vapor content (Fig. 8) of $w = 11.0$ kg/m$^2$ according to the 06 UTC-radiosonde. The range corrected signals of the CHX-2 ceilometer from midnight to noon are shown in Fig. 13 to illustrate the aerosol stratification of that day. The top of the aerosol layer was slowly decreasing from 2.3 km at midnight to 1.75 km at 09 UTC. Then convection led to a rapid increase

of the mixing layer again. Compared to Case A its vertical extent of the aerosol layer was larger. The validation range was set to $0.75 \leq z \leq 1.55$ km.



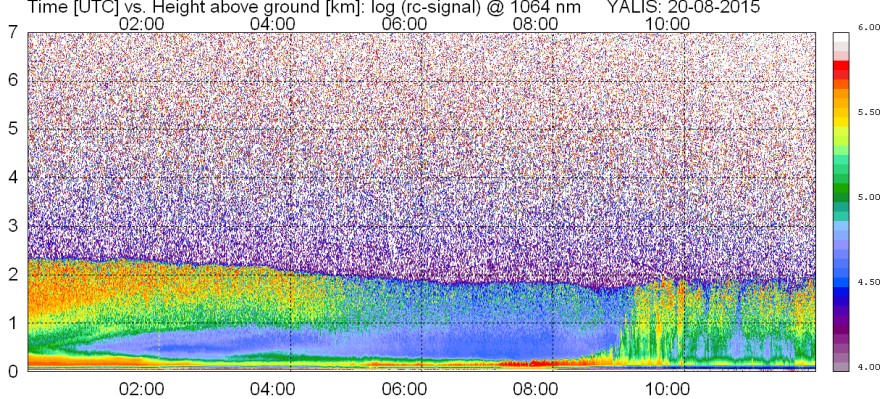

**Figure 13.** Time-height cross section of the range corrected signal (in arbitrary units, logarithmic scale) of the CHX-2 from 20 August 2015 from midnight until noon (including Case B). Time is given in UTC, and the height above ground in km; note, that the maximum height shown is not the full measurement range of the ceilometer.

The water vapor number density is shown in Fig. 14a: the black line indicates the profile of the relative humidity of the 06 UTC radiosonde (launched at 04:47 UTC) whereas the red lines show the number density $n_w$ (06 UTC and 12 UTC as solid and dashed lines, respectively). A very sharp decrease of $n_w$ at 2.0 km can be found which is in perfect agreement with the top of the aerosol layer (Fig. 13 at 05 UTC). The transmission of the particles $T_p$ at 1064 nm is derived from RALPH measurements as

described for Case A. As can be seen in Fig. 14b it is comparable with Case A (see Fig. 10b). This is plausible from AERONET measurements of the aerosol optical depth $\tau_p$: at 500 nm $\tau_p$ exhibits sort of a temporary minimum with $\tau_p = 0.11$ and was thus only slightly larger than during Case A ($\tau_p = 0.10$); the day before and later the same day $\tau_p$ was considerably larger. This is also plausible from visual inspection of Fig. 13. The water vapor transmission $T_{w,\mathrm{eff}}$ for different wavelengths is larger than in Case A as the water vapor concentration was lower. The spectral dependence of $T_{w,\mathrm{eff}}$ (see Fig. 14c) is the same as in Case A

with maximum values at 905 nm and minimum values at 915 nm.

The Angström exponent was derived from AERONET Level 2.0 data between 04:56 UTC and 11:38 UTC. From averaging 25 retrievals we found $\kappa_w = 1.10 \pm 0.14$, almost identical to $\kappa_n$ but smaller than $\kappa_a = 1.30$. Thus, we assume a range of $0.96 \leq \kappa \leq 1.24$, resulting in $1.162 \leq L_p \leq 1.214$. The $\kappa(z)$-profile determined from the RALPH signals at 532 nm and 1064 nm shows an increase with height within the validation range from $\kappa = 0.92$ to $\kappa = 1.15$, which is in good agreement with the mean

AERONET value.

With this input the conversion function $\eta$ is calculated according to Eq. (11). The results are shown in Fig. 15 – similar to Fig. 11 – for 905 nm (red) and 915 nm (blue). The dashed lines concern the constant $\kappa$ assumption with $\kappa = 0.96$ (short dashed) and $\kappa = 1.24$ (long dashed) as the range of uncertainty of $\kappa$. The solid lines shows the conversion factor $\eta$ in the case of the height-dependent Angström exponent. The absolute values of the conversion functions $\eta$ are similar to Case A but the the





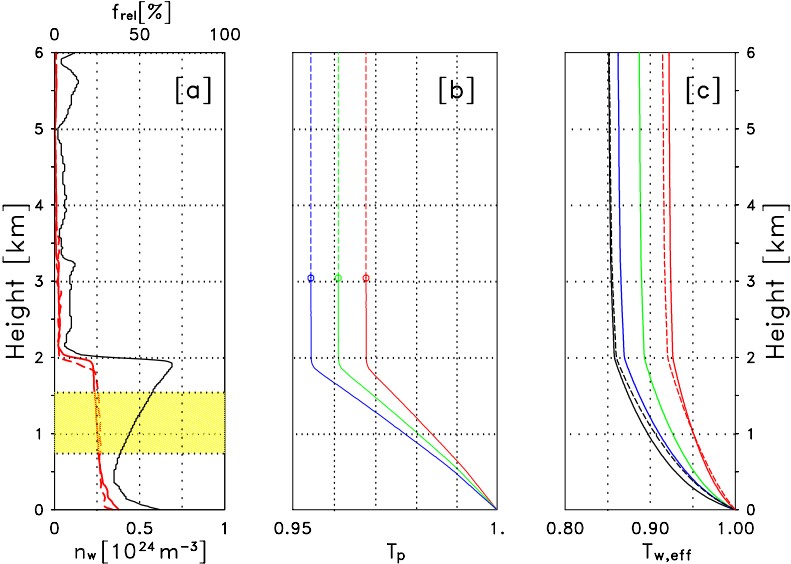

**Figure 14.** (a): Profile of the relative humidity (black line) in percent, see labels at the top, and profiles of the water vapor number density (in $10^{24}$ molecules/m$^3$, labels at the bottom) for the 06 UTC radiosonde ascent (solid red line) and the 12 UTC ascent (dashed red line) of 20 August 2015. The validation range is highlighted in yellow. (b): Particle transmission $T_p$ at 1064 nm derived from the Klett inversion of averaged RALPH signals (20 August 2015, 05 UTC – 08 UTC, Case B) assuming a lidar ratio of $S_p = 45$ sr (red), $S_p = 55$ sr (green), and $S_p = 65$ sr (blue), respectively. The circles indicate the reference height. (c): Effective water vapor transmission $T_{w,eff}$ for different laser wavelengths $\lambda_{on}$ (solid lines: 900 nm (black), 905 nm (red), 907 nm (green), 910 nm (blue); dashed lines: 915 nm (black), 925 nm (red)); analogously to Fig. 10.

height dependence is quite different as expected from the radiosonde profiles (Fig. 10a and Fig. 14a). Again, the $S_p$-dependence is negligible.

Having determined $\eta$ the validation is done analogously to Case A with the key parameters summarized in Table 5. Fig. 16 shows the wavelength dependence of the slope $dF/dz$ for the CL51-1 (black solid line) and CL51-2 (red solid line) assuming a height-dependent $\kappa(z)$ and with the range due to the uncertainty of the lidar ratio indicated by the dashed lines of the same color. The best agreement is found for $\lambda_{on} = 915$ nm ($dF/dz = -2.2 \cdot 10^{-3}$ km$^{-1}$, $\Delta F = 0.7$ %) in the case of CL51-1, and for $\lambda_{on} = 915$ nm ($dF/dz = -1.3 \cdot 10^{-2}$ km$^{-1}$, $\Delta F = 0.8$ %) in the case of CL51-2. The dependence on $S_p$ is negligible as was the case in Case A. The absolute values of $dF/dz$ are again much smaller than the corresponding values for 1064 nm ($dF/dz = -0.12$ km$^{-1}$ and $-0.13$ km$^{-1}$). The wavelength of the best agreement for CL51-2 is the same for Case A and Case B, however, this is solely a consequence of the criterion ($\|dF/dz\| = $ min). According to Fig. 16 any wavelength in the range of strong absorption leads to a good agreement. For the CL51-1 we find a wavelength in the same range, whereas a





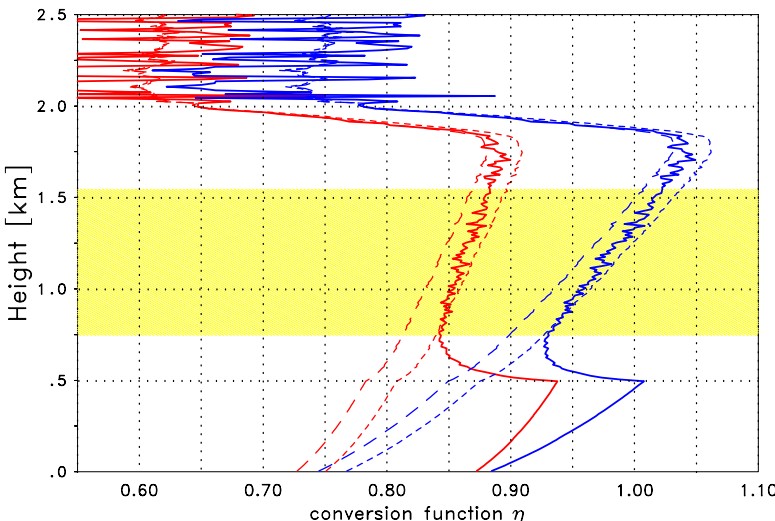

**Figure 15.** Analogously to Fig. 11: conversion function $\eta$ at 905 nm (red) and 915 nm (blue) for Case B. The dashed lines are for height-independent $\kappa = 0.96$ (short-dashed), and for $\kappa = 1.24$ (long-dashed). The solid line is for the height-dependent $\kappa(z)$. The validation range (in yellow) was set to 0.75 km and 1.75 km.

wavelength in the moderate part of the absorption band (918 nm) was found in Case A. The reasons must remain unclear. One can suspect that it is an effect of the changing temperature of the CL51-1: it was between 31 °C and 28 °C for Case A, whereas it was between 25 °C and 30 °C for Case B. In contrast the temperature of the CL51-2 has changed less. Though the actual shift of the wavelength cannot be retrieved by this kind of investigations due to the ambiguity of the effective absorption, the

temperature change is not sufficient to fully explain the different $\lambda_{\mathrm{on}}$ if we assume the 0.27 nm $\mathrm{K}^{-1}$ dependence (as specified by the manufacturer, see Wiegner and Gasteiger (2015)) as the only influencing factor.

If a constant $\kappa$ is used for the calculation of $\eta$, the slopes $dF/dz$ are even smaller as obvious from the green (CL51-1) and blue (CL51-2) curves. In both cases we can find wavelengths yielding a perfect agreement with $dF/dz = 0$.

The good agreement of the range corrected signals – ceilometer measurements vs. extrapolated reference measurements –

is confirmed by their slope $s$: from Eq. (17) we can expect that $1.09 < s < 1.15$ considering the uncertainties of the different contributions, whereas we get from the measurements of the CL51-ceilometers $s = 1.10$ and $s=1.11$, respectively, i.e. an even better agreement than in Case A. For the CL31-ceilometers we find again larger $s$-values (1.37 and 1.24); they correspond to a too strong decrease of the signals to be explained by water absorption only. For Case B the slope of the range signals of the two CS135 ceilometers is slightly smaller ($s = 1.08$ and $s = 1.04$, respectively) but quite close to the expected range, and very

small slopes $dF/dz$. For the CS-1 even a perfect agrement can be found at 908 nm and 918 nm. For the specified emission



**Table 5.** Key parameters of the validation for Case B. The minimum slope $dF/dz$ for an integer wavelength is given, or $dF/dz = 0$ if the corresponding curve shown in Fig. 12 crosses the zero-line (for an non-integer wavelength). According to Eq. (17) the decrease of the range corrected signal $s$ should be $1.09 < s < 1.15$

| Ceilometer | $dF/dz$ | $s$ |
|------------|---------|------|
| CL51-1 | $-2.2$E-3 | 1.10 |
| CL51-2 | $-1.3$E-2 | 1.11 |
| CL31-1 | $-2.8$E-1 | 1.37 |
| CL31-2 | $-1.5$E-1 | 1.24 |
| CS-1 | 0 | 1.08 |
| CS-2 | $-8.9$E-3 | 1.04 |

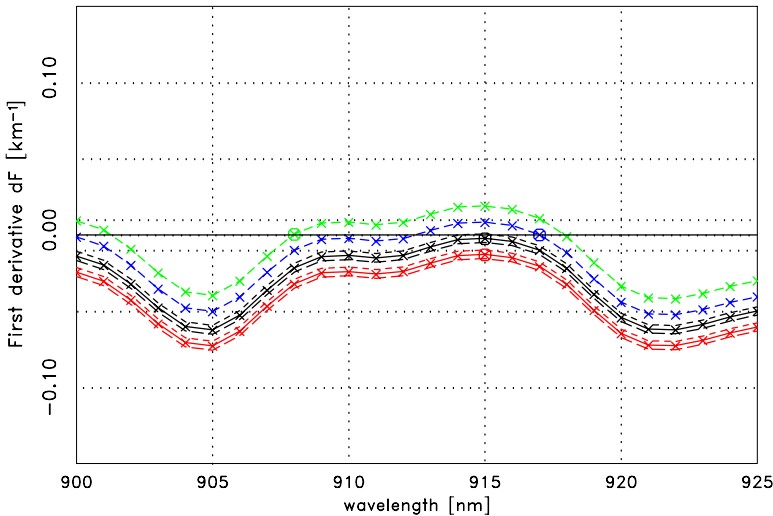

**Figure 16.** Same as Fig. 12, but for 20 August 2015, 05 UTC – 08 UTC (Case B).

wavelength of 912 nm we find $dF/dz = 0.012$. If, however, the validation range is extended to 1.75 km, the validation is not successful, suggesting deteriorated CS135-signals in the uppermost part of the mixing layer. This is not the case for the CL51 ceilometers.





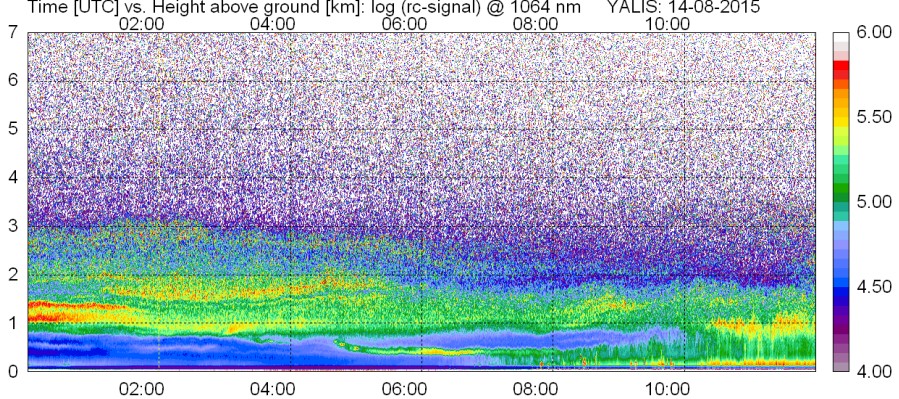

**Figure 17.** Same as Fig. 9 but for 14 August 2015 (Case C).

### 4.5.3 Case C: 14 August 2015

The third case concerns 14 August 2015 with the time period from 00 UTC to 03 UTC. The total water vapor content with $w$ = 33.0 kg/m$^2$ according to the 00 UTC-radiosonde was quite large (see Fig. 8). The overview of the aerosol distribution from midnight to noon based on the range corrected signal of the CHX-2 ceilometer is shown in Fig. 17. Elevated aerosol layers
between approximately 0.8 km and 3.0 km persisting for several hours after midnight are the dominant feature. The optical depth at 500 nm – averaged over 6 hours in the morning – was $\tau_p = 0.36$, which is well above the average.

The Angström exponent was found to be $\kappa = 1.55 \pm 0.014$ when averaging 18 AERONET retrievals between 04:47 UTC and 07:11 UTC. Compared to the previous cases $\kappa$ was quite large and the variability very small. It perfectly agrees with the Angström exponent derived from $\beta_p$ at 532 nm and 1064 nm: RALPH retrievals assuming $S_p = 55$ sr show an almost constant
$\kappa(z)$ with $\kappa = 1.57$ in an altitude of 1.1 km and $\kappa = 1.62$ in 2.7 km. Note, that the uncertainty of $\kappa$ due to the uncertainty of $S_p$ is however comparable large in this case. With the typical assumption of $\pm 10$ sr for the uncertainty of $S_p$ we get an uncertainty of $\kappa$ of $\pm 0.1$ and $\pm 0.05$ at the lower and upper boundary of the validation range. The validation range (yellow area in Fig. 18) was selected as $1.1 < z < 2.7$ km.

The conversion function $\eta$ is calculated as before. The resulting profiles are shown in Fig. 18. Again, the red lines correspond
to 905 nm whereas the blue lines are for 915 nm. The solid lines are for the height-dependent $\kappa$, the dashed lines for the constant $\kappa$ as derived from AERONET. According to the quite similar $\kappa$-values differences of $\eta$ are almost negligible. Note, that the $\eta$-values are significantly larger than in the previous cases with less atmospheric water vapor.

Results of the validation using extrapolated RALPH signals and CL51 ceilometer measurements are shown in Fig. 19. A survey of the key parameters is provided by Table 6. Assuming the height-dependent $\kappa(z)$ slopes $dF/dz = 0$ can be found for
both CL51 ceilometers. The values of $\|dF/dz\|$ are very small, and many integer wavelengths can be found that show slopes close to zero. Compared to the other examples $\Delta F$ is slightly larger (1.4 %). If the AERONET-based $\kappa$ is used (green and





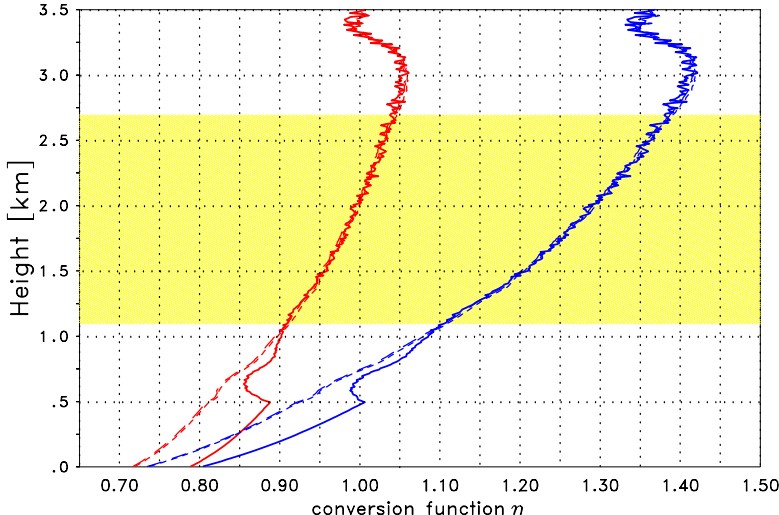

**Figure 18.** Analogously to Fig. 11: conversion function $\eta$ at 905 nm (red), and 915 nm (blue). The solid lines are for $\kappa(z)$ as derived from the RALPH measurements, the short dashed line is for a constant $\kappa = 1.53$, the long-dashed lines for $\kappa = 1.56$. The validation range is between 1.1 km and 2.7 km (yellow area). Measurements are from 14 August 2015, 00 UTC – 03 UTC (Case C).

**Table 6.** Key parameters of the validation for Case C. The minimum slope $dF/dz$ for an integer wavelength is given, or $dF/dz = 0$ if the corresponding curve shown in Fig. 12 crosses the zero-line (for an non-integer wavelength). According to Eq. (17) the decrease of the range corrected signal $s$ should be $1.46 < s < 1.72$

| Ceilometer | $dF/dz$ | $s$ |
|---|---:|---|
| CL51-1 | 0 | 1.64 |
| CL51-2 | 0 | 1.67 |
| CL31-1 | −1.5E-1 | 2.20 |
| CL31-2 | 1.9E-2 | 1.50 |
| CS-1 | 1.5E-1 | 1.18 |
| CS-2 | 2.2E-1 | 1.06 |





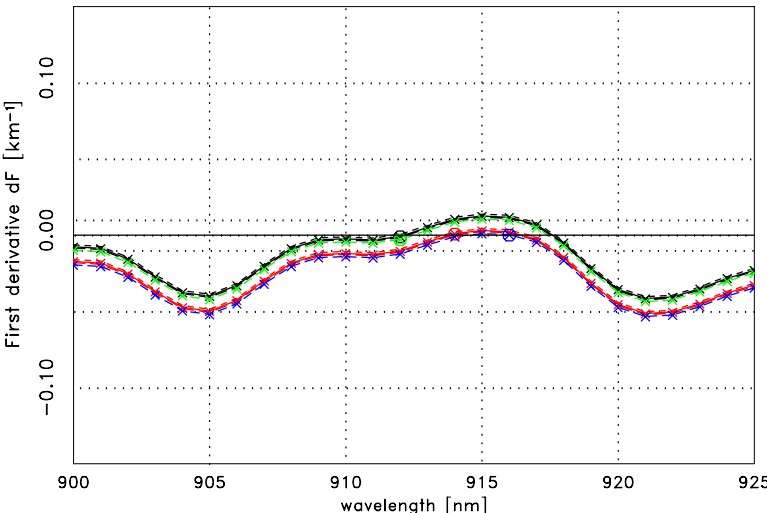

**Figure 19.** Same as Fig. 12, but for 14 August 2015, 00 UTC – 03 UTC (Case C).

blue line), perfect agreement is also found as expected from the similarity of the Angström exponents. The good agreement of the measured and extrapolated signals is confirmed by their slope $s$: we find that $1.46 < s < 1.72$ considering the inherent uncertainties of the individual contributions to Eq. (17). The values derived from the measurements of the CL51-ceilometers are $s = 1.64$ and $s = 1.67$, respectively, and fall very well into the expected range. For the other ceilometers the water vapor

validation is acceptable only for the CL31-2. The quite different results for the two CL31 ceilometers shows that obviously difference occur even if the same type of ceilometer is evaluated.

## 5   Summary and conclusions

The large number of ceilometers and the fact that they can be run unattended and fully automated makes them potentially very attractive for aerosol observations. Consequently several attempts have been made to use them for aerosol remote sensing –

though this does not comply with the intended use of the manufacturers. By exploiting ceilometer data in depth one became aware of the role of water vapor absorption and its influence on the retrieval of particle optical properties. Approaches to correct for this effect have been proposed recently (Wiegner and Gasteiger, 2015), however, a validation was still missing.

To assess the ceilometers' potential in a quantitative way field campaigns were set up to compare them with reference lidar systems, to investigate their long term stability and their operability in different environments. A corresponding activity was

conducted in summer 2015 in Lindenberg, Germany, in the framework of the CeiLinEx2015-campaign. One of the scientific





objectives was the above mentioned validation of retrieving aerosol optical properties in the case of water vapor absorption. The multi-wavelength Raman lidar RALPH served as reference. The focus of this paper is on two types of Vaisala ceilometers (CL51 and CL31) and the CS135 of Campbell Scientific, all operating in the spectral range around 910 nm where water vapor absorption is significant.

Validation was performed on the basis of comparing backscatter signals. We extrapolate the reference signal from 1064 nm to the wavelengths of the ceilometers for validation, and exploit the ratio of both. The validation was considered successful if a height-independent ratio could be found for any wavelength in the specified range of the emission, note, that the actual wavelength is not exactly known. For this purpose the spectral dependence of particle optical properties has to be known; we use information either from co-incident AERONET data or from the inversion of RALPH backscatter signals.

It turns out that the spectral extrapolation and the selection of the validation range are the most crucial points of the validation. In particular we recommend that the vertical range used for the validation should be selected very carefully; typically it is limited to the upper part of the mixing layer. The reason is that on the one hand the range of incomplete overlap cannot be corrected with sufficient accuracy, on the other hand that the ceilometer signals above the mixing layer are either too noisy or substantially influenced by signal artefacts. Consequently different validation range might apply for each ceilometer.

It could be demonstrated that the water vapor correction was successful in the case of the CL51 ceilometers. In the case of the CL31 and CS135 ceilometers the validation was not always successful: though the agreement between the measured signals and the extrapolated reference signal was better with than without correction, the agreement was in general, but not always, worse in comparison to the CL51. In particular for the CS135 no generally valid conclusions could be found.

We conclude from the measurements during CeiLinEx2015 that at the present state-of-the-art of ceilometers correction for
water vapor absorption to improve aerosol remote sensing seems to be reasonable for Vaisala CL51 ceilometers. For the other ceilometers participating in CeiLinEx2015 further studies are required as other error sources – not known in detail yet – seem to dominate the water vapor effect. Anyway, in all cases uncertainties remain as long as the emitted spectrum of the laser is unknown.

If in the future manufacturers of automated lidars and ceilometers aim at quantitative retrievals of aerosol optical properties,
either the emitted wavelength should be monitored or wavelengths influenced by gaseous absorption should be avoided. In this context an investigation of the benefit of radiation at 808 nm as applied by the Cimel CE372 lidar (Ancellet et al., 2018) would be interesting. Then, a key problem influencing the water vapor correction can be solved. Further steps forward are expected from an additional characterization of the hardware, monitoring meta data of relevant system parameters, and regular dark measurements to be able to correct for signal artefacts. As a consequence an optimized adaptation of the validation range
might be possible. All suggestions would certainly help to improve future validation activities. Moreover, correction of artefacts may extend the range where quantitative $\beta_p(z)$-profiles can be provided. We think that it is worthwhile to go in this direction as it offers a lot of new applications, e.g. the combination of passive and active remote sensing for aerosol retrievals (e.g., Román et al., 2018).





*Acknowledgements.* The CeiLinEx2015 measurement campaign received support for organization and analysis of results from the European Cooperation in Science and Technology (COST; www.cost.eu) Action ES1303 "TOPROF". J. A. Bravo-Aranda has received funding from the Marie Sklodowska-Curie Action Cofund 2016 EU project – Athenea3i under grant agreement No. 754446. J. Gasteiger has received funding from the European Research Council (ERC) under the European Union's Horizon 2020 research and innovation programme (Grant
5  No. 640458, A-LIFE).

We are grateful to Marc-Antoine Drouin (LMD, France) for developing pre-processing software for the CeiLinEx2015 data. Moreover, we want to thank Robert Begbie (DWD, Germany) and the staff at the Meteorological Observatory (DWD) in Lindenberg for the organization of CeiLinEx2015, technical support and maintenance of the ceilometers.

## 6   Data availability

10  Data of CeiLinEx2015 campaign are available from: Pattantyús-Ábrahám, M., Mattis, I., Begbie, R., Bravo-Aranda, J. A., Brettle, M., Cermak, J., Marc-Antoine Drouin, Geiß, A., Görsdorf, U., Haefele, A., Haeffelin, M., Hervo, M., Komínková, K., Leinweber, R., Münkel, C., Pönitz, K., Vande Hey, J., Wagner, F., and Wiegner, M.: The Dataset of the CeiLinEx2015 Ceilometer-Inter-comparison Experiment, Version v001, doi: 10.5676/DWD/CEILINEX2015, 2017





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
