# Peer review of "Aerosol backscatter profiles from ceilometers: validation of water vapor correction in the framework of CeiLinEx2015"

_Atmospheric Measurement Techniques, 2018_

## Referee Comment (RC1) · Li (Referee) · 16 Dec 2018

**Li (Referee)**

ccli@pku.edu.cn

This study performed the validation about the ceilometers whose spectral range around 910 nm are influenced by water vapor absorption when used to retrieval aerosols. The authors compared ceilometer backscatter signals with measurements of the reference system extrapolated to the water vapor regime. They solved the key problem of the spectral extrapolation of particle optical properties. They found the vertical range where validation is possible is limited to the upper part of the mixing layer due incomplete overlap, and the in general low signal to noise ratio and signal artefacts above that layer. A quite good agreement between the extrapolated reference signals and the

measurements in the case of CL51 ceilometers at one or more wavelengths in the specified range of the laser diode's emission was obtained after the effective water vapor correction. The paper is clearly written. Overall, this is a nice paper and well-conceived effort of value to the community of atmospheric measurements. I suggest its publication on the journal of AMT after some technical corrections revised by the authors. No extra data or processing are needed. From a general aspect, I would suggest to shorten the paper a bit, for example "The validation range" and "the results parts".

Technical corrections:

1, Page 3, Line 4, "measurements of a CYY-2B ceilometer (CAMA) that was deployed in China were reported", here, the reference is missing.

2, Some legends should be added on the figures, for examples, Figure 1 - 7, 10-12, 14-16, 18 and 19, to describe the colored lines or circles for being easily understood, instead of only describe in the texts.

3, It is not necessary to show the low signal to noise ratio parts in the high free atmosphere on Figure 9, 13, and 17. My suggestion is to shorten the vertical height to 3.5 or 4.0 km to present the images more clearly.

---

## Referee Comment (RC2) · Zhang (Referee) · 24 Dec 2018

General comments:

The manuscript presents an evaluation of water vapor correction for ceilometers operating in the spectral range around 910 nm. The author solved important issues and provided a clear correction process. The accuracy of aerosol profile inversion from ceilometers data can be improved significantly. The paper is well written, validation of water vapor correction is discussed in extensive detail. The paper is clearly relevant and appropriate for publication to AMT. I recommend the paper for publication after technical corrections.

Technical corrections:

Online wavelength may drift with temperature, as mentioned at Page28, Line5. CL31/51 have a wide bandwidth receiving optical filter (36nm@50% pass band according to CL31/51 manual) to adapt to wavelength drift within the operating range of -40 to 60 degreeC. In order to be perfect, I would suggest that the author should give the temperature adaptation range of the correction for the ceilometer without temperature control.

---

## Author Comment (AC1) · 10 Jan 2019

We thank Chengcai Li and Tianshu Zhang for reviewing our manuscript and their useful comments. Due to Christmas vacations and New Year our replies are delayed by approximately two weeks.

We have included our replies to each of their comments and attached the revised version of the manuscript for tracking the changes we made. They are highlighted as displayed by latexdiff. Our replies to each comment of the reviewers are given below (in italics).

[Figure]

**Replies to Chengcai Li's comments**

[...] The paper is clearly written. Overall, this is a nice paper and well conceived effort of value to the community of atmospheric measurements. I suggest its publication on the journal of AMT after some technical corrections revised by the authors. No extra data or processing are needed. From a general aspect, I would suggest to shorten the paper a bit, for example "The validation range" and "the results parts".

→ *Thanks for the positive overall evaluation. According to the suggestion we have gone through the paper and have shortened or deleted several sentences whenever it seems that they are not necessary to understand the text. A few typos have been corrected as well. The changes are highlighted in the attached revision of the paper.*

Page 3, Line 4, "measurements of a CYY-2B ceilometer (CAMA) that was deployed in China were reported", here, the reference is missing.

→ *These measurements are reported by Liu et al. (2018). This reference is already given on page 3, line 16. As no other references could be found in peer-reviewed journals and because that paper summarizes the technical specifications of the instrument we have added the same reference again.*

Some legends should be added on the figures, for examples, Figure 1 - 7, 10-12, 14-16, 18 and 19, to describe the colored lines or circles for being easily understood, instead of only describe in the texts.

→ *We agree with the reviewer's suggestion that the lines, colors and symbols should be found without the need to scroll through the paper. For this reason we had included all relevant definitions in the caption of the corresponding figure. That*

*leads indeed to a certain degree of complexity and could be confusing. So we have added legends whenever sufficient space was left in the figures and overloading could be avoided. The following changes were made:*

*Figs. 1, 2, 5, 6, 7, 12, 15, 16, 18: legends were added.*

*Figs. 10 and 14: in the left panel the color of lower labels was changed from black to red (so it is more intuitive how labels and lines belongs together), and a legend was added to the panel in the center.*

*Fig. 11: the colors were changed to have a clear relationship to the wavelength. A legend was added.*

*Fig. 19: a legend was added and the vertical scale was changed to better see the differences of the curves.*

*Figs. 9, 13, 17: changed as described below.*

*Figs. 3, 4 were left unchanged as we believe that they are already quite concise.*

It is not necessary to show the low signal to noise ratio parts in the high free atmosphere on Figure 9, 13, and 17. My suggestion is to shorten the vertical height to 3.5 or 4.0 km to present the images more clearly.

$\rightarrow$ *We had the same intent as the reviewer: for this reason we did not show the full vertical range (i.e. $\approx$ 15 km) but only 7 km. To enhance the visibility of the internal structure of the aerosol layers we have used a color code that is based on the histogram of $Pr^2$-values; each color represents the same number of "pixels". Doing this, the structure is pronounced and the image "looks nice" as the contrast to the free (almost aerosol free) troposphere is obvious. This gets lost if we drastically reduce the vertical range to 3.5 km or 4.0 km. As a compromise we applied 5 km (actually 5.1 km for technical reasons) as upper limit of Figs. 9, 13 and 17 in the revised version. We are confident that this is acceptable for the reviewer.*

**Replies to Tianshu Zhang's comments**

[...] The paper is well written, validation of water vapor correction is discussed in extensive detail. The paper is clearly relevant and appropriate for publication to AMT. I recommend the paper for publication after technical corrections.

→ *Again we appreciate the positive rating of our study.*

Online wavelength may drift with temperature, as mentioned at Page28, Line5. CL31/51 have a wide bandwidth receiving optical filter (36nm@50% pass band according to CL31/51 manual) to adapt to wavelength drift within the operating range of -40 to 60 degree C. In order to be perfect, I would suggest that the author should give the temperature adaptation range of the correction for the ceilometer without temperature control.

→ *The reviewer is right. This is indeed an important detail that should we reported in the manuscript, in particular, as this is different to the Campbell ceilometers. Accordingly we have included this information in Section 2, see page 4 of the revised version.*

---

## Author Comment (AC3)

[revised manuscript text omitted]
_{\mathrm{on}}, z) = \frac{P(\lambda_{\mathrm{off}}, z)}{\eta(z)} := P_{\mathrm{extra}}(\lambda_{\mathrm{on}}, z) \tag{14}$$

The term $P_{\mathrm{extra}}(\lambda_{\mathrm{on}}, z)$ is introduced to make clear that it is not a measurement but a signal extrapolated to $\lambda_{\mathrm{on}}$. For a quantitative assessment of the agreement between $P_{\mathrm{extra}}(\lambda_{\mathrm{on}}, z)$ and the measured ceilometer signal $P_{\mathrm{ceilo}}(z)$ at an actually unknown wavelength in the "water vapor regime", we define the ratio $F$ as

$$F(\lambda_{\mathrm{on}}, z) = c_{\mathrm{norm}} \, \frac{P_{\mathrm{ceilo}}(z)}{P_{\mathrm{extra}}(\lambda_{\mathrm{on}}, z)} \qquad \text{with} \quad c_{\mathrm{
[revised manuscript text omitted]

---

## Author Response (AR2)

We were informed by the associate editor that a third reviewer has send comments to the original manuscript after our submission of the revised version. Our replies to these comments are included below. Note, that most of the reviewer's concerns are related to each other, so there is also some small "overlap" in our replies. We think we have answered all questions of the reviewer in detail, including referencing to our manuscript we his/her concerns are discussed and explained. When it was necessary we have extended the revised version of our manuscript, but we have kept it short as we wanted to avoid redundancy.

The revision of the revised version as a "latexdiff-file" is attached.

**Replies to the third reviewer**

(1) All figures are lacking of legends. It is thus very challenging to read through this manuscript. (I see now you already corrected for this according to previous comments)

> → *This was indeed a severe shortcoming. Now we included legends. See revised version of the manuscript.*

(2) The authors define the minimum of the absolute value of the slope $dF/dz$ as the criterion for a correct treatment of the water vapor absorption. Should not the maximum of the absolute value be used? Under this circumstance, a small value of $dF/dz$ indicates a correct treatment of the water vapor absorption?

> → *The function $F$ is the ratio of the measured signal and the reference signal corrected for water vapor, see Eq. (15). If the water vapor correction is perfect the signals should look the same (i.e., $F = const.$), and consequently the derivative $dF/dz$ should be zero. In other words, as the slope of a ceilometer signal is proportional to the transmission, it means that the absorption is correctly considered. This is in agreement with the last sentence of reviewer comment. To a certain degree we also considered the absolute value of the slope when we introduced $s$, see Eq. (17). This parameter is the decrease of the signal in the validation range. It was shown that it sometimes was too large indicating that the decrease of the signal was too strong: this can only be explained by shortcomings of the instruments as otherwise an unknown atmospheric absorber must have been present.*

(3) A figure showing the curve of $dF/dz$ should be given to better illustrate the meanings of $dF/dz$. From my point of view, the value of $\Delta F$ seems to be more appropriate for the criterion for a correct treatment of the water vapor absorption.

→ *We believe that the best (most intuitive) illustration is provided by Fig. 2. Here the reader directly sees that the decrease of the CL51-signals is very close to that of the extrapolated (water vapor corrected) reference signal, that the decrease of the CL31-signals is stronger, and the decrease of the CS135-signal strongest. This means that the CL31- and CS135-signals are stronger attenuated with increasing height than can be explained by physical reasons. So the slope of the signals can be used to indicate whether the water vapor correction is correct.*

*The requested curves of $dF/dz$ are already shown e.g. in Fig. 16, as a function of the wavelength. We assume that the reviewer additionally wants to have information on the vertical structure of $F$ as this is the basis for $dF/dz$. Thus, one example of $F$ as a function of height is shown below. The black and red curves correspond to the CL51-1 and CL51-2 measurements, respectively. One can see that the slope $dF/dz$ is virtually zero and $F = 1$ as constrained by the normalization defined in Eq. (15). Results shown are found for $\lambda_0 = 908$ nm (green circle in Fig. 16) and 917 nm (blue circle in Fig. 16) in consistence with the manuscript. The mean $dF/dz$ within the validation range is also indicated in Fig. 16.*

*The value of $\Delta F$ indicates how large the deviations of $F$ from unity are. Here we treat $F = 0.99$ and $F = 1.01$ as an deviation of 1% (see dotted vertical line). The larger the deviations are the larger the "fluctuations" of $F$ within the validation range. This helps to understand how noisy the ceilometer signals are, and thus helps to select a reasonable validation range: in the lowermost part the ratio is not realistic due to the problems in the overlap range (see detailed discussion in our manuscript), in the uppermost part it suffers from noise. Note, that $\Delta F$ alone would not be good criterion as it could be small even in case that $F$ shows a strong decrease/increase with height (i.e., water vapor correction has failed).*

*We hope that this example and our explanations show that all relevant information contained in this figure is also available from the manuscript. Moreover, the manuscript is already quite long (see reviewer #1) so we don't want to include more figures (different cases,*

*different ceilometers, different wavelengths).*

[Figure]

Figure 1: An example of $F$ for CL51-1 (black line) and CL51-2 (red line), $\kappa$ = const. (Case B). Compare to Fig. 16 of the manuscript and see detailed explanations above.

(4) "For the other ceilometers the water vapor validation is acceptable only for the CL31-2. The quite different results for the two CL31 ceilometers shows that obviously difference occur even if the same type of ceilometer is evaluated." Why could the result be different even for the same type ceilometer? What would be the potential reason for this?

→ *It was one purpose of our study to show the "real world". As we have mentioned in the introduction and (after the revision) in the conclusions, the ceilometers are primarily designed to determine the cloud base height, and to be installed as networks. Thus, these instruments are adequate if they meet this requirement and if they are "inexpensive". The manufacturers act accordingly, e.g. they use components that fulfill the requirements but not necessarily more than that. As an example, the wavelength is not monitored as this is not an issue for*

*cloud detection. In other words: if there are differences in the same type of ceilometers nobody cared (in the past) because the primary purpose (cloud detection, layer detection) is not concerned.*

*Recently atmospheric scientists investigated the potential for more sophisticated applications (aerosol remote sensing). In that framework additional requirements were formulated; it is now up to the manufacturers if they want to upgrade their systems (e.g., increase the stability, monitor the wavelength, provide temperature control [if not yet included], reduce/eliminate artefacts) or not. To a certain extent the decision will be economically driven. The goal of our and similar studies is to explain what will be possible (from a researcher's point of view) if certain (new) features will be implemented in the instruments.*

*We have added a clarifying sentence to the end of Section 4.5.3 of the manuscript.*

(5) In the conclusion section, "the CS135 no generally valid conclusions could be found". Could the authors comment more about this? Does this mean that the CS135 ceilometer does not operate properly?

→ *No, this does not mean that the CS135 does not operate properly. Failure of the water vapor correction does not prevent the detection of cloud base heights (see previous comment for additional details).*

*We have added a summarizing sentence to the conclusions: "These findings however do not question the primary purpose of all ceilometers, the ability to determine cloud base height."*

(6) "Anyway, in all cases uncertainties remain as long as the emitted spectrum of the laser is unknown." How much is the contribution of the wavelength variation of the laser to the uncertainties?

→ *The differences can be seen in Figs. 12, 16 and 19: the slope of the signals change with wavelength (as the absorption is wavelength-dependent). Note, that at the current state it does not play a role whether the best agreement is at one wavelength or another (within the possible range of the emission), as it cannot be verified (that led us to our recommendation for a wavelength-monitor); it is only relevant that there is one (or more) wavelength in the range allowing a successful water vapor correction. It has been demonstrated in our paper that*

*this is the case for some ceilometers but not for all. As mentioned in the manuscript, a careful selection of the validation range is important as well.*

*To make this clear we have again pointed out in the conclusions of the revised version: "...
[revised manuscript text omitted]